# Riverine carbon export in the arid-semiarid Wuding River catchment on the Chinese Loess Plateau

Lishan Ran[1], Mingyang Tian[2], Nufang Fang[3], Suiji Wang[4], Xixi Lu[2,5], Xiankun Yang[6], Frankie Cho[1]

[1]Department of Geography, The University of Hong Kong, Pokfulam Road, Hong Kong
[2]School of Ecology and Environment, Inner Mongolia University, Hohhot, China
[3]State Key Laboratory of Soil Erosion and Dryland Farming on the Loess Plateau, Institute of Soil and Water Conservation, Northwest A&F University, Yangling, China
[4]Institute of Geographical Sciences and Natural Resources Research, Chinese Academy of Sciences, Beijing, China
[5]Department of Geography, National University of Singapore, Singapore
[6]School of Geographical Sciences, Guangzhou University, Guangzhou, China

**Abstract:** Riverine export of terrestrially derived carbon represents a key component of the global carbon cycle. In this study we quantify the fate of riverine carbon within the Wuding River catchment on the Chinese Loess Plateau. Export of dissolved organic and inorganic carbon (DOC and DIC) exhibited pronounced spatial and temporal variability. While DOC concentration first presented a downward trend along the river course and then increased in the mainstem river, it showed no significant seasonal differences and was not sensitive to flow dynamics. This likely reflects the predominance of groundwater input over the entire year and its highly stable DOC. DIC concentration in the loess subcatchment is significantly higher than that in the sandy subcatchment, due largely to dissolution of carbonates that are abundant in loess. In addition, bulk particulate organic carbon content (POC%) showed strong seasonal variability with low values in the wet season owing to input of deeper soils by gully erosion. The downstream carbon flux was $(7.0\pm1.9)\times10^{10}$ g C year$^{-1}$ and dominated by DIC and POC. Total $CO_2$ emissions from water surface were $(3.7\pm0.6)\times10^{10}$ g C year$^{-1}$. Radiocarbon analysis revealed that the degassed $CO_2$ was 810–1890 years old, indicating the release of old carbon previously stored in soil horizons. Riverine carbon export in the Wuding River catchment has been greatly modified by check dams. Our estimate shows that carbon burial through sediment storage was $(7.8\pm4.1)\times10^{10}$ g C year$^{-1}$, representing 42% of the total riverine carbon export from terrestrial ecosystems on an annual basis $((18.5\pm4.5)\times10^{10}$ g C year$^{-1})$. Moreover, the riverine carbon export accounted for 16% of the catchment's net ecosystem production (NEP). It appears that a significant fraction of terrestrial NEP in this arid-semiarid catchment is laterally transported from the terrestrial biosphere to the drainage network.

## 1. Introduction

Rivers play an exceptionally significant role in the global carbon cycle by directly linking terrestrial ecosystems and the oceans (Cole et al., 2007;Regnier et al., 2013;Drake et al., 2017). Prior studies indicate that the amount of terrestrially derived carbon entering inland waters was substantially larger than that discharged into the oceans mainly through fluvial transport of global rivers (Mendonça et al., 2017;Battin et al., 2009). With respect to river systems, this carbon imbalance suggests that rivers are not passive pipes simply transporting terrestrial carbon,

but are biogeochemically active in processing massive quantities of carbon along the river course. Riverine carbon is subject to a number of physical and biogeochemical processes such as burial, evasion, *in situ* production, and decomposition. The $CO_2$ emissions from water surface of global rivers and streams combined are conservatively estimated at 0.65–3.2 Pg C year$^{-1}$ (Lauerwald et al., 2015;Drake et al., 2017;Raymond et al., 2013). In addition, carbon loss due to long-term sediment storage through burial is also substantial, ranging from 0.15 to 0.6 Pg C year$^{-1}$ (Battin et al., 2009;Cole et al., 2007;Mendonça et al., 2017;Clow et al., 2015). Inclusion of $CO_2$ emissions and carbon burial in sediments is thus critical for a holistic understanding of carbon cycling in river systems at different spatial scales.

Although studies on riverine fluxes of carbon have been considerably increasing over the last 20 years, great uncertainties remain to be properly resolved even for catchment-scale assessments, not to mention the larger regional and global estimates (Marx et al., 2017). An important source for these uncertainties is the underrepresentation of current carbon flux measurements, which are mostly confined to tropical and boreal rivers that are sensitive to climate change. In contrast, few studies have investigated the terrestrial and fluvial fluxes of carbon in arid and semiarid rivers though they are globally abundant (Tranvik et al., 2009). Increased concerns over global riverine carbon export and emissions necessitate an improved understanding of carbon cycling in these underexplored rivers. Studying their riverine carbon cycling on the basis of individual catchments will shed light on refining global riverine carbon flux estimates and thereby assessing their biogeochemical importance, as has been done for tropical and temperate rivers (e.g., Butman and Raymond, 2011; Richey et al., 2002).

With the role of arid-semiarid rivers in global riverine carbon cycle in mind, we investigated the transport and fate of carbon from terrestrial ecosystems through the drainage network to the catchment outlet in the medium-sized Wuding River catchment on the arid-semiarid Loess Plateau (northern China). The overall aim of this study was to quantify the fate of riverine carbon among its three pathways; that is 1) downstream export to catchment outlet, 2) $CO_2$ evasion from water surface, and 3) organic carbon (OC) burial through sediment storage within the arid-semiarid Wuding River catchment. To achieve this aim, a catchment-scale carbon balance was constructed. The major objectives are to: 1) explore the spatial and temporal variability of riverine carbon export, 2) trace the sources and age of the emitted $CO_2$ using carbon isotope techniques, and 3) evaluate the riverine carbon cycle in relation to the catchment's terrestrial ecosystem production. This study is built upon our earlier work of Ran et al. (2017) in which we analyzed environmental controls and dam impacts on riverine $CO_2$ emissions. These results will provide insights into riverine carbon studies for rivers in arid-semiarid climates and improve the accuracy of extrapolation from watershed-based carbon studies to global-scale estimates.

**2. Study area and methods**
**2.1 Study area**
The Wuding River (37–39$^o$ N; 108–110.5$^o$ E) is one of the largest tributaries of the Yellow River and is located on the central Chinese Loess Plateau (Fig. 1). Its drainage area is 30,261 km$^2$ and mean water discharge during the period 1956–2017 is 35 m$^3$ s$^{-1}$ or 11.2×10$^8$ m$^3$ year$^{-1}$. Based on geomorphological landscape, the catchment can be further divided into the southeastern loess

subcatchment, generally covered with 50–100 m deep loess soils, and the northwestern sandy subcatchment composed mainly of aeolian sand (Fig. 1). While grassland is extensive in the sandy subcatchment, agriculture and grassland are the primary land use types in the loess

subcatchment with traditional ploughing tillage as the dominant land management practice. Annual precipitation during the period 1956–2004 decreases from 500 mm in the southeast to 300 mm in the northwest, of which 75% falls in the wet season from June until September (Li et al., 2007). Several heavy storms in summer can account for half of the annual precipitation. Except the periods of heavy storms, hydrological regime is controlled by groundwater input,

especially in the sandy subcatchment (Li et al., 2007). Due to highly erodible loess and sparse vegetation, the Wuding River catchment has experienced a maximum, decadal averaged soil erosion rate as high as 7000 t $km^{-2}$ $year^{-1}$ during the period 1956–1969 (Ran et al., 2017).

Check dams have long been proposed as an effective soil conservation strategy. By 2011, more

than 11,000 check dams have been constructed (Ran et al., 2017). Because their primary purpose is for reducing sediment loss, these structures are generally designed without sluice gates. Consequently, most of the sediment from upstream hillslopes and gullies can be effectively trapped (Ran et al., 2013), resulting in a short life time for these dams because of rapid sediment accumulation, generally less than 20 years (Xu et al., 2013). The resulting organic carbon (OC)

burial is likely substantial, but remains to be quantified (Zhang et al., 2016). Because of widespread presence of calcite in loess (up to 20%; Zhang et al., 1995) and carbonate dissolution and precipitation under dry climate, this catchment shows hard-water attributes in rivers and check dam-formed reservoirs featuring high dissolved solids. Its mean alkalinity was 3850 µmol $L^{-1}$ and long-term river water $CO_2$ partial pressure ($pCO_2$) ranged between 1000 and 2500 µatm

(Ran et al., 2015a).

## 2.2 Field sampling and laboratory analysis
While detailed information has been provided in Ran et al. (2017), a brief description is provided here. Three sampling campaigns were conducted in the Wuding River catchment in 2015: before

the wet season (March–April; denoted as spring), during the wet season (July–August; summer), and after the wet season (September–October; autumn). Sampling was not performed in winter due to ice coverage. The sampling was performed at 74 sites, including 60 river sites in six Strahler order rivers (Strahler, 1957) and 14 reservoir sites in 8 check dam-formed reservoirs (Fig. 1). Moreover, monthly samples were collected at the catchment outlet Baijiachuan gauge

(Fig. 1) in 2017 and daily hydrological records for 2015 and 2017 were also retrieved from the gauge. The sampling frequency was intensified (i.e., 2-h intervals) during typical flood events.

We employed the drifting floating chamber technique to measure *in situ* $CO_2$ emissions (Ran et al., 2017). Briefly, an infrared Li-7000 gas analyzer (Li-Cor, Inc, USA) was connected to a

rectangular chamber (volume: 17.8 L) via rubber-polymer tubes to measure $CO_2$ concentration changes inside the chamber over time. We also measured *in situ* surface water $pCO_2$ using the headspace equilibrium method by means of the Li-7000 gas analyzer (Müller et al., 2015). Triplicate measurements at each site showed a high consistency with 3% variability only. Finally, surface water $pCO_2$ was calculated and calibrated with the solubility constants for $CO_2$

from Weiss (1974). To determine the age of the emitted $CO_2$, we collected samples for [14]C
analysis by using the precipitation method widely used in groundwater dating studies (Vita-Finzi
and Leaney, 2006). After the $CO_2$ emissions measurement, the accumulated $CO_2$ in the chamber
was directly injected into 50 mL $SrCl_2$ solution in a closed recirculating loop using an external
pump. Reaction of chamber $CO_2$ with $SrCl_2$ results in the precipitation of $SrCO_3$. The

precipitated $SrCO_3$ was then filtered, dried, and stored in a cool and dark environment until
analysis. Eleven $SrCO_3$ samples were collected at four sites during the three campaigns.

Water samples for dissolved organic and inorganic carbon (DOC and DIC) were filtered on site
shortly after collection using Whatman filters (0.45 µm pore size). DOC was analyzed on an

Elementar Vario TOC select analyzer following procedures in Ran et al. (2017). Triplicate
injections indicated an analytical precision of less than 3%, and the average of the three injection
results was calculated to represent the DOC concentration. Total alkalinity was determined by
triplicate end-point titrations in the field with 0.1 M HCl and methyl orange indicator. DIC was
calculated from total alkalinity, pH, and temperature by using the program $CO_2$calc (Robbins et

al., 2010). Both DOC and DIC data have been presented in Ran et al. (2017). We also drilled
sediment cores within 4 check dams by using a soil auger (Fig. 1). Sediment samples were
collected at 20-cm intervals and the drilling depth was 4–6 m depending on sedimentation
history. Samples collected from filters and sediment coring for particulate organic carbon (POC)
were first dried for 12 h and then pulverized using a mortar and pestle. The obtained fine powder

was fumigated by concentrated HCl at 65 °C for 24 h to remove inorganic carbon and measured
using a PerkinElmer 2400 Series II CHNS/O elemental analyzer (analytical error: <0.3%).
Isotopic signature of the eleven $SrCO_3$ samples was determined using accelerator mass
spectrometry (AMS) at the Beta Analytic Radiocarbon Dating Laboratory (Miami, USA). The
[14]C results were reported as percent modern carbon (pMC) based on modern standard and

conventional radiocarbon ages (year before present, BP) were calculated using the [14]C half-life
(5568 years) following the procedures outlined by Stuiver and Polach (1977). Meanwhile, stable
carbon isotope ($\delta^{13}C$) was measured using an isotope ratio mass spectrometer (IRMS) and its
values were reported in ‰ relative to the VPDB standard at a precision of ±0.3‰ or better.

**2.3 Carbon fluxes and $CO_2$ emissions**
Using the monthly sampling results of DOC and DIC concentrations in water and bulk POC
content (POC%) of the total suspended sediments (dry weight) measured at the catchment outlet
Baijiachuan gauge, we calculated the yearly DOC, DIC, and POC fluxes from the Wuding River
catchment. Because daily flow and sediment records are available, the yearly carbon flux was

calculated by using the Beale's stratified ratio estimator which generally exhibits greater
estimation accuracy and lower bias than other flux estimation techniques (Lee et al., 2016). The
estimator can be expressed as follows:

$$\mu_y = \mu_x \frac{m_y}{m_x} \left( \frac{1 + \frac{1}{n} \frac{S_{xy}}{m_x m_y}}{1 + \frac{1}{n} \frac{S_x^2}{m_x^2}} \right) \qquad (1)$$

where $\mu_y$ is the estimated flux, $\mu_x$ is the mean daily water discharge for the year measured, $m_y$ is

the mean daily carbon flux for the days on which the dissolved and particulate carbon
concentrations were determined, $m_x$ is the mean daily water discharge for the days on which the

carbon concentrations were determined, and $n$ is the number of days on which the carbon concentrations were determined. Furthermore,

$$S_{xy} = \frac{1}{(n-1)} \sum_{i=1}^{n} x_i y_i - n m_x m_y \qquad (2)$$

and

$$S_x^2 = \frac{1}{(n-1)} \sum_{i=1}^{n} x_i^2 - n m_x^2 \qquad (3)$$

where $x_i$ is the individual measured discharge, $y_i$ is the daily carbon flux for each day on which the dissolved and particulate carbon concentrations were measured. Clearly, the yearly DOC, DIC, and POC fluxes are derived from $m_y/m_x$, which is defined as the ratio of the mean of measured fluxes to the mean of water discharge of the days when fluxes were quantified. This ratio is used with the overall mean water discharge ($\mu_x$) to estimate the annual carbon flux. The calculated annual fluxes of DOC, DIC, and POC were then added up to determine the total downstream carbon export from the Wuding River catchment.

Areal fluxes of $CO_2$ emissions across water-air interface ($F_{CO_2}$, mmol m$^{-2}$ d$^{-1}$) were determined from the slope of the linear regression of $pCO_2$ against time (r$^2$ ≥0.97):

$$F_{CO_2} = 1000 \times (\frac{dpCO_2}{dt})(\frac{V}{RTS}) \qquad (4)$$

where, $dpCO_2/dt$ is the slope of $CO_2$ change within the chamber (Pa d$^{-1}$; converted from µatm min$^{-1}$), $V$ is the chamber volume, $R$ is the gas constant, $T$ is chamber temperature (K), and $S$ is the area of the chamber covering the water surface (0.09 m$^2$). Particularly, results of the areal $CO_2$ emissions have been presented in our earlier work (Ran et al., 2017).

Total OC burial behind check dams was estimated by multiplying annual sediment deposition rate by POC% in sediments. Our earlier work (Ran et al., 2013) has estimated the average annual sediment deposition rate behind all check dams in the study catchment by considering sediment input into each check dam and its sediment trapping efficiency. To calculate $CO_2$ efflux from the entire catchment, we estimated the areal extent of river water surface by using the 90-m resolution Shuttle Radar Topography Mission (SRTM) DEM data set (Ran et al., 2015b). A threshold value of 100 cells was first set to delineate the drainage network on the assumption that a stream initiates within the cells. The delineated network was then classified using the Strahler ordering system (Strahler, 1957). Because the width of all rivers is less than the resolution and it fluctuates between dry and wet seasons, we measured widths of all sampled rivers and aggregated them based on stream order to calculate the water surface area. For reservoirs, our earlier work (Ran and Lu, 2012) has identified their location and areal extent. Both the delineated and reservoirs were calibrated through ground truthing. We further assumed that each round of field sampling is representative of $CO_2$ emissions for equally four months (i.e., spring sampling: January–April (120 d); summer sampling: May–August (123 d); autumn sampling: September–December (122 d)). With this assumption in mind, we calculated the first-order estimate of yearly $CO_2$ efflux from both rivers and reservoirs.

### 2.4 Estimation of terrestrial ecosystem production
To further evaluate the riverine carbon export, we compared the total carbon entering the drainage network with the catchment's net ecosystem production (NEP). MOD17A3H

(MODIS/Terra Net Primary Production) produced by USGS (https://lpdaac.usgs.gov/) was used
to first estimate net primary productivity (NPP). The MOD17A3H Version 6 provides global
NPP estimates at 500-m pixel resolution and in units of kg C $m^{-2}$. While NPP is an important
indicator of carbon uptake by terrestrial ecosystems, it does not account for carbon losses
through heterotrophic soil respiration ($R_h$). Heterotrophic soil respiration due to heterotrophs
tends to release a significant fraction of the sequestered carbon into the atmosphere, depending
on soil temperature, moisture, and substrate availability (Wei et al., 2015). Therefore, the NEP
was used for the assessment and it can be estimated by subtracting $R_h$ from NPP:

$$\text{NEP} = \text{NPP} - R_h \tag{5}$$

To calculate $R_h$, total soil respiration ($S_R$) was first derived from the global soil $CO_2$ efflux
database described by Raich and Potter (1995) who estimated $S_R$ at a 0.5º latitude by longitude
spatial scale. $S_R$ was then divided into its two components of autotrophic and heterotrophic soil
respiration. $R_h$ was finally estimated according to the assumption by Hanson et al. (2000) that $R_h$
accounts for 54% and 40% of $S_R$ in forested and non-forested areas, respectively.

## 3. Results
### 3.1 Lateral riverine carbon fluxes

DOC concentrations ranged from 1.4 to 9.5 mg $L^{-1}$ in the three sampling seasons with both the
lowest and highest values observed in spring. The DOC averaged 5.0±1.6, 5.2±1.3, and 4.5±1.6
mg $L^{-1}$ in spring, summer, and autumn, respectively, without discernible seasonal variation in
both the loess and sandy subcatchments. Although statistically insignificant, DOC first exhibited
a downward trend along the river course and then increased in the 6th order mainstem river in
both subcatchments (Fig. 2). While the DOC in the headwater 1st−2nd order streams (4.7−5.4
mg $L^{-1}$) was on average 9−21% higher than in the 3th−5th order streams (4.2−4.9 mg $L^{-1}$), it
increased to 5.2−6.1 mg $L^{-1}$ in the 6th order mainstem river, representing an increase of 18−36%
relative to the 3th−5th order streams. The POC% varied from 0.28% to 1.72% and spatially
remained largely constant from the headwater stream to the mainstem (Table S1 in Supplement).
However, it showed pronounced seasonal variations. The average POC% in spring, summer, and
autumn was 0.91±0.32%, 0.44±0.10%, and 0.69±0.21%, respectively.

With the pH in the range of 7.68−9.29, the calculated DIC was approximately equal to alkalinity.
The Wuding waters presented significantly higher DIC than DOC concentrations. DIC in spring,
summer, and autumn varied in the range of 39−119, 32−132, and 34−143 mg $L^{-1}$ with the
average at 62.1±21.4, 66.7±23.8, and 67.7±21.9 mg $L^{-1}$, respectively. In the loess subcatchment,
the DIC declined remarkably from headwater streams towards the mainstem river (one-way
ANOVA test, p≤0.05; Fig. 3a); but it remained constant in the sandy subcatchment from the 1st
order through the 5th order streams (Fig. 3b). The high DIC values in the 6th order mainstem
river in the sandy subcatchment (Fig. 3b) is reflective of the confluence of the two
subcatchments. If only the 1st−5th order streams were considered, DIC in the sandy
subcatchment was 38% lower than that in the loess subcatchment.

At Baijiachuan gauge, the DIC remained highly stable at 39.0±4.7 mg $L^{-1}$. The DOC
concentrations were 16% higher in the wet season than in the dry season while the POC%

(range: 0.15−1.16%) in the former was less than half of that in the latter. The mean DOC and POC% were 3.3±0.4 mg L$^{-1}$ and 0.61±0.23%, respectively (Table S2 in Supplement). The flow regime in 2017 was significantly biased by an extreme flood in July (rainfall of 203 mm and spontaneous discharge of 4490 m$^3$ s$^{-1}$; see He et al. (2018) and Fig. S1 in Supplement) with the precipitation ~26% higher than the long-term average. Hence, we used the hydrological data for 2015, which is 4% lower than the long-term average, to calculate downstream carbon export by assuming that carbon concentration was comparable in 2015 and 2017. The annual downstream carbon export at this gauge was estimated to be $(7.0±1.8)×10^{10}$ g C, of which the DIC, DOC, and POC fluxes were $(3.0±0.4)×10^{10}$, $(0.3±0.03)×10^{10}$, and $(3.7±1.8)×10^{10}$ g C, respectively. DOC flux was around 10% of the DIC and POC fluxes, comprising only 4% of the total flux. DIC and POC fluxes were comparable, accounting for 53% and 43%, respectively, of the total flux.

**3.2 CO$_2$ emissions from rivers and check dam-formed reservoirs**

In our earlier work, we calculated the areal CO$_2$ emissions from rivers (Ran et al., 2017). In the sandy subcatchment, the mean CO$_2$ efflux from the 1st order headwater streams to the 6th order mainstem river was 280, 422, 155, 216, 256, and 238 mmol m$^{-2}$ d$^{-1}$, respectively. In the loess subcatchment, it was 70, 78, 80, 57, 209, 268 mmol m$^{-2}$ d$^{-1}$, respectively. In association with the water surface area over the three seasons (Table S4 in Supplement), total CO$_2$ emissions in 2015 were $(3.65±0.5)×10^{10}$ g C, of which 42% was degassed from the sandy subcatchment rivers and 58% from the loess subcatchment rivers. At the catchment scale, CO$_2$ outgassing along fluvial transport first decreased from upland headwater rivers until the 4th order rivers, and then increased remarkably in the 5th and 6th order rivers in both subcatchments (Fig. 4a). The headwater 1st and 2nd order rivers accounted for 26% of the total CO$_2$ efflux (Fig. 4b). With the biggest areal extent of water-air interface (43% of the total; Table S4 in Supplement), the 6th order mainstem river contributed 54% of the total CO$_2$ efflux (Fig. 4b).

CO$_2$ effluxes from check dam-formed reservoirs varied from -23.5 to 66.5 mmol m$^{-2}$ d$^{-1}$ in spring, -33.5 to 19 mmol m$^{-2}$ d$^{-1}$ in summer, and -17 to 42.1 mmol m$^{-2}$ d$^{-1}$ in autumn. The mean CO$_2$ efflux for these three seasons was 4.2, -16.2, and 12.3 mmol m$^{-2}$ d$^{-1}$, respectively (Ran et al., 2017). Of the 8 reservoirs, 2 reservoirs are located in the sandy subcatchment and 6 in the loess subcatchment (Fig. 1). Reservoir CO$_2$ effluxes in the sandy subcatchment were constantly higher or less negative than that in the loess subcatchment with the mean efflux at 10.4 and -2.9 mmol m$^{-2}$ d$^{-1}$, respectively. Currently, there are 337 reservoirs with the water surface varying from 0.01 to 10.35 km$^2$ (Fig. S2 in Supplement). Total water surface area is 107 km$^2$, including 31.8 km$^2$ in the sandy subcatchment and 75.2 km$^2$ in the loess subcatchment. Assuming the water surface area remained constant (i.e., no significant seasonal fluctuations), the spring and autumn CO$_2$ effluxes were summed to 246 million mol and the summer CO$_2$ efflux was -208 million mol (Table 1). These added up to an annual net CO$_2$ efflux of 38±280 million mol (or $0.05×10^{10}$ g C), which is statistically indistinguishable from zero due largely to the spatial variation between the sandy and loess subcatchment reservoirs in spring (Table 1). When added with the river efflux estimate, the catchment total CO$_2$ efflux was $(3.7±0.6)×10^{10}$ g C in the year 2015, of which the reservoir CO$_2$ efflux accounted for less than 1.4%.

The isotopic composition of the emitted $CO_2$ varied significantly between sampling sites and between seasons (Table 2). The sandy subcatchment (site S1; Fig. 1) showed the most depleted $\delta^{13}C$ signature (-30.2‰). With the $\delta^{13}C$ values most depleted in spring, the mean $\delta^{13}C$ values in spring, summer, and autumn were -30.2±3.2‰, -24.5±5.6‰, and -23.2±2.3‰, respectively. The $\Delta^{14}C$ values also displayed seasonal variations and the radiocarbon age ranged from 810 to 1890

years (Table 2; Fig. 5). The emitted $CO_2$ exhibited the oldest age in spring at all the 4 sites with the age in summer and autumn 36% and 29% younger, respectively. The average $^{14}C$ age in the three seasons was 1610, 1038, and 1140 $^{14}C$ year BP, respectively. There was no discernible correlation between DIC and DOC concentrations and the isotopic composition.

### 3.3 OC burial behind check dams

Based on our earlier estimate of sediment trapping, the trapping efficiency in this catchment is 94.3% and total sediment deposition rate is $3720\times10^{10}$ g year$^{-1}$ (Ran et al., 2013). Analysis of sediment profiles from the four check dams (Fig. 1) shows the POC% varied from 0.1% to 0.5% with high POC% values in the surface sediments (0−60 cm) and it declined rapidly with depth

and remained constant thereafter at around 0.2% (Fig. 6; Table S3 in Supplement). The mean POC% was 0.21±0.11%. Total OC burial behind check dams was estimated to be $(7.8±4.1)\times10^{10}$ g C year$^{-1}$.

### 3.4 Terrestrial NPP and NEP fluxes

The NPP in the Wuding River catchment in 2015 was spatially heterogeneous (Fig. 7). The mean areal NPP was 221 g C m$^{-2}$ and the total NPP was $(668±60)\times10^{10}$ g C. Based on the global soil respiration flux database (Raich and Potter, 1995), the $S_R$ for this catchment is the range of 400−500 g C m$^{-2}$ year$^{-1}$. Hence, we used 450±50 g C m$^{-2}$ year$^{-1}$ to represent its soil respiration. This rate is consistent with recent measurements under different vegetation types in this arid-

semiarid region (e.g., Fu et al., 2013; Jia et al., 2013). Recent land use studies show that forest cover in this catchment occupies only 5% of the total area (Wang et al., 2014), while the remaining is dominated by cropland or dry grassland (Li et al., 2007). Using the ratios of autotrophic to heterotrophic soil respiration for forested and non-forested land suggested by Hanson et al. (2000), $R_h$ was estimated to be 183±20 g C m$^{-2}$ year$^{-1}$. By subtracting $R_h$ from

NPP, a first-order estimation shows a NEP of 38±28 g C m$^{-2}$ year$^{-1}$ or $(114±85)\times10^{10}$ g C year$^{-1}$ for the entire catchment. The NEP represented only 17% of the NPP, and heterotrophic soil respiration consumed 83% of the sequestered carbon.

### 4. Discussion

**4.1 Carbon export dynamics within the catchment**
Carbon export from terrestrial ecosystems into drainage networks is controlled by hydrological regime, geomorphological landscape, biogeochemical processes, and human impact within the catchment of concern (Noacco et al., 2017;Stimson et al., 2017). For the Wuding River catchment, its DOC concentrations are comparable to the global average DOC of 5.4 mg L$^{-1}$

while its POC% is lower than most rivers in the world (mean: 0.95%; Ludwig et al., 1996). Stream water OC is susceptible to degradation by microbial reactions during transit (Raymond et al., 2016). The downstream DOC decline in the 1st−5th order streams likely suggests the

mineralization of the bioavailable fraction of DOC along the river course (Fig. 2), especially in spring and autumn. This can also be seen from the 9–21% higher DOC concentrations in the

headwater 1st–2nd order streams than in the 3rd–5th order streams. This mineralization is generally associated with increasing water residence time for bacterial respiration in downstream streams due to longer travel times which increase the potential for in-stream processes on DOC. In contrast, the deeply incised headwater streams in the Wuding River catchment exhibit an opposite landscape with the flow velocities increasing from headwater streams to the mainstem

river (Ran et al., 2017). Thus, the decreasing water residence time cannot fully explain the decreasing DOC concentration. Instead, the gradually increasing temperature with declining elevation might have enhanced bacterial respiration (Peierls and Paerl, 2010). The water temperature in the lowland streams was on average 2–5 $^{o}$C higher than in the headwater streams (Ran et al., 2017).


The high DOC values in the 6th mainstem river reflect direct DOC influxes from low-order streams (Fig. 1) and the mixture of carbon from the two subcatchments. Owing to the insignificant seasonal difference in DOC concentration measured across the catchment, there was no discernible relationship between DOC and flow based on the spatial sampling results

($p>0.05$). Although the extensive implementation of agricultural tillage practices in April and May tends to mobilize vast amounts of OC, carbon export through surface runoff into the drainage network is limited to episodic high-discharge events in June to September. The timing inconsistency suggests that the mobilized soil OC in this dry catchment was either adsorbed within deeper soils or released into the atmosphere after mineralization. Lateral export into the

drainage network by surface runoff is negligible. The predominance of groundwater input over the entire year and its highly stable DOC illustrate the insensitivity of DOC concentration to flow dynamics. In contrast, the spatial heterogeneity of DIC with higher values in the loess subcatchment was likely caused by dissolution of carbonates which are abundant in loess (Zhang et al., 1995).


The POC% in suspended sediments in the Wuding River catchment is at the lower end of global rivers (range: 0.3%–10.1%; Ludwig et al., 1996), which likely reflects the contribution of ancient sedimentary OC of ~0.5% to POC in fluvial sediments (Meybeck, 1993). This can also be seen from the isotopic signature of the Yellow River sediment that is primarily derived from the Loess

Plateau, especially the studied Wuding River and other nearby rivers. By using carbon isotope techniques, Wang et al. (2012) discovered that the exported POC is quite old (4110–8040 [14]C year BP) and is largely derived from highly weathered loess soils and ancient kerogen. The much lower POC% in summer than in spring and autumn reflected the impact of gully erosion, which is quite common on the Loess Plateau during heavy rainstorm periods (Wang et al., 2017). Gully

erosion is usually associated with the mobilization of sedimentary rocks that generally have a substantially lower POC% (i.e., 0.2–0.3%; Zhang et al., 1995; Ran et al., 2015a) than the surface soils. As a result, input of sedimentary rocks into rivers caused the lower POC% in summer, thereby generating a negative correlation between POC% and sediment concentration.

With respect to $CO_2$ outgassing, the higher effluxes in the drier sandy subcatchment reflect the stronger impact of groundwater input, although both sub-catchments are heavily controlled by groundwater inflow. While several heavy rainstorms in summer are responsible for a large share of the annual precipitation (i.e., >70%; Wang et al., 2017), our field measurements in 2015 did not capture the storm-caused $CO_2$ outgassing. Thus, the $CO_2$ emissions results reveal largely the
groundwater-derived $CO_2$ degassing. This may have caused considerable uncertainty in the annual $CO_2$ outgassing estimation (see discussion below). Although the sandy subcatchment rivers exhibited higher areal $CO_2$ effluxes than that in the loess subcatchment in all the 1st−5th order rivers except the 6th order mainstem river, the lower contribution of $CO_2$ emissions from the former (42%) is because its water surface accounts for 32% only of the total water surface. In
comparison, the larger contribution of the loess subcatchment rivers (58%) reflects their higher drainage density and larger water surface area (68% of the total; Table S4 in Supplement).

Unlike natural rivers showing strong $CO_2$ outgassing, the measured reservoirs presented considerably lower and even negative $CO_2$ effluxes. The contrasting magnitude and direction of
$CO_2$ exchange suggest the physical and biogeochemical differences between lotic and lentic waters. Compared with rivers with fast moving water and high sediment concentrations, reservoirs display greatly reduced flow turbulence and enhanced algal production resulting from increased light penetration after the settling of suspended sediment (Cole et al., 2007). Analysis of chlorophyll-*a* also shows that it is 100% higher in reservoirs than in rivers in summer and
autumn (Ran et al., 2017), indicative of carbon uptake by phytoplankton through photosynthesis. In the sandy subcatchment, the predominance of groundwater with high $pCO_2$ has probably maintained its relatively higher reservoir $CO_2$ effluxes (mean: 10.4 mmol $m^{-2}$ $d^{-1}$). For the loess subcatchment reservoirs, intensive nutrient loading from agricultural fields may have facilitated the growth of phytoplankton like algae, causing the net carbon uptake (mean: -2.9 mmol $m^{-2}$
$d^{-1}$). Overall, these reservoirs differ from their tropical counterparts that typically act as strong $CO_2$ source hot spots (Barros et al., 2011;Deemer et al., 2016), yet they are consistent with other temperate reservoirs with similar landscape attributes (Knoll et al., 2013). Given the global abundance of hard-water reservoirs and their unique carbon processing mechanisms (Tranvik et al., 2009), estimating global $CO_2$ emissions from reservoirs must pay comparable attention to
these currently underrepresented reservoirs as to their tropical counterparts.

**4.2 Downstream carbon export at catchment outlet and OC burial**
The monthly carbon export at Baijiachuan gauge illustrates diverse responses of different carbon species to hydrological regime. Hydrologic storm events in wet seasons play a disproportionately
important role in transporting terrestrially-derived carbon. Our high-frequency sampling during flooding periods at Baijiachuan gauge indicates that DOC concentrations were 26% higher in the flooding periods than that in normal flow conditions. The positive correlation between DOC export and hydrography demonstrates the enhanced leaching of organic matter from surface vegetation and organic-rich top soil layers (Hernes et al., 2008). Moreover, increased stream
velocities in the flooding periods have reduced water residence time and consequently, even the bioavailable fraction of DOC could be quickly transported downstream, resulting in a greater export of DOC (Raymond et al., 2016). Clearly, this positive response contradicts the indiscernible relationship between DOC and flow discharge within the catchment. This is

probably because the three intensive seasonal samplings did not capture the carbon export in
high-flow conditions. The flow discharge during the three sampling periods varied in the range of $0.002–105$ $m^3$ $s^{-1}$, which largely reflects the carbon export processes during low flow to, at most, medium flow conditions. In comparison, the high-frequency sampling at Baijiachuan gauge captured the carbon export during extremely high flows ($200–1760$ $m^3$ $s^{-1}$, Table S2 in Supplement). In addition, the DIC concentration displayed a weak sensitivity to flow dynamics.
Widespread presence of calcite in loess and intensive carbonate dissolution tend to provide sufficient DIC input, which have probably prevented the dilution effect observed in many other rivers (Ran et al., 2015a;Raymond and Cole, 2003).

The substantially lower POC% values in the wet season may have reflected the hydrodynamic
sorting of terrestrially derived organic carbon. Recent studies on size distribution of POC% in the Yellow River (the Loess Plateau) suggest that 85% of its POC is concentrated in sediments with grain size smaller than 32 µm (Zhang et al., 2013;Wang et al., 2012). Coarser sediments transported by high discharges in the wet season thus have a lower POC%. In addition, the lower POC% is likely associated with the erosion processes as discussed earlier. With respect to
sediment sources on the Loess Plateau, it has been widely realized that more than 50% of the sediment in wet seasons, especially during heavy rainstorm periods, is derived from gully erosion (Wang et al., 2017;Ran et al., 2015a). Mobilization of deeper soils with a low POC% (i.e., 0.2–0.3%) and subsequent fluvial transport resulted in the observed low POC% values in the wet season. Our results of 0.15–0.26% for samples collected during floods agreed well with
the low carbon content in deeper soils. Despite the low POC%, however, the POC flux in the wet season is considerable on an annual basis because of the high sediment loading, accounting for 65% of the annual total POC flux.

$CO_2$ outgassing during flooding periods have also been significantly enhanced due largely to
stronger near-surface turbulence and thus a higher gas transfer velocity (Fig. 8). The average $CO_2$ efflux for the monitored flooding period was 5 times that in normal flow conditions (196 vs. 39 mmol $m^{-2}$ $d^{-1}$). When looking at the annual total fluxes, episodic high-discharge events were responsible for a significant percentage of annual carbon export though the duration of high-discharge events made up 4% only of the sampling year 2017. A conservative calculation using
the sampling results at Baijiachuan gauge indicates that 85% of the annual downstream carbon export occurred during the three extreme floods (Fig. S1 in Supplement). Therefore, any sampling strategies missing episodic high-discharge events would create great uncertainties for annual-scale carbon export estimates (Lee et al., 2017;Jung et al., 2014). This is particularly true for arid-semiarid catchments, such as the Wuding River studied here, where episodic rainfall
events make an exceptionally large share of annual water and sediment export.

The decreasing POC% in the deposited sediments with depth demonstrates the OC burial efficiency. Soil OC within the Wuding River catchment is spatially homogeneous. The content in hillslope soils varies from 0.4–0.7% and it is less than 0.2% in the gully soils due to strong
mineralization in the Quaternary loess (Wang et al., 2017), which is roughly equal to the POC% in the trapped sediments. The negligible POC% difference likely reflects the spatial location and

the high sediment trapping efficiency of check dams. Most check dams are located at the bottom of highly erodible loess gullies. This spatial closeness to erosional sites suggests that the eroded soils can be rapidly deposited after a short delivery distance (Wang et al., 2011). In addition, the

distinctive behaviour of soil erosion in the study catchment can partially explain the small POC% difference. Recent studies indicate that, if the rainstorm intensity is sufficiently strong, all grain-size fractions of loess soils on hillslopes can be eroded without sorting (Zheng et al., 2008). Based on combined use of $^{137}$Cs and $\delta^{13}$C techniques, Wang et al. (2017) discovered that approximately 70% of the eroded soil OC can be buried by check dams in the Wuding River

catchment. However, it is worth noting that the POC% showed significant variations with depth (Fig. 6). The estimated total OC burial rate is associated with uncertainty and warrants further investigation by considering POC% changes with depth and other secondary OC sources (e.g., phytoplankton).

In view of the huge sediment deposition behind check dams, the resulting OC burial represents an important carbon sink for the atmosphere that would have otherwise been partially mineralized to form $CO_2$ or $CH_4$ in the water column and outgassed along fluvial delivery (Drake et al., 2017;Battin et al., 2009). It is important to recognize that, as a top priority soil conservation strategy, numerous check dams have been constructed on Loess Plateau over the

past 60 years and more are under construction to replace the filled ones (Zhang et al., 2016;Wang et al., 2017). Assessing the potential OC burial efficiency and amount may have important implications for regional and even global carbon budgets. Regional estimates of OC burial in lakes have recently been made (Zhang et al., 2017;Kastowski et al., 2011). Considering the larger number of check dams and reservoirs of China, quantifying their OC burial will be critical

for a more robust OC burial assessment in global lakes and reservoirs (Mendonça et al., 2017).

**4.3 Carbon isotopic signature in the emitted $CO_2$**

$CO_2$ emissions from rivers originate from decomposition of organic matter derived from terrestrial ecosystems and/or aquatic photosynthesis. The emitted $CO_2$ exhibited a $^{13}$C-depleted

$\delta^{13}$C signature significantly different from that originated from carbonate-dominant rivers (i.e., 0‰, Brunet et al., 2009). As stated earlier, widespread carbonate dissolution in the Wuding River catchment is the primary source of DIC in its groundwater (Zhang et al., 1995;Chen et al., 2005). Although we did not analyze the $\delta^{13}$C signature of DIC, prior studies suggest that it generally ranges from -6.7‰ to -12.9‰ in Loess Plateau rivers, indicative of strong dominance

of carbonate dissolution (Liu and Xing, 2012). For natural rivers with the DIC dominated by $HCO_3^-$, kinetic isotope fractionation due to $CO_2$ outgassing tends to enrich the $\delta^{13}$C of DIC by 3–5‰ (Doctor et al., 2008). Therefore, the emitted $CO_2$ is less likely to be derived from the interactions between water and carbonates, because the kinetic isotope fractionation process is not able to compensate the great discrepancy in $\delta^{13}$C. This is consistent with the $\delta^{13}$C changes in

soil $CO_2$ in sandy catchments (Gillon et al., 2012).

Instead, the $\delta^{13}$C values of the emitted $CO_2$ are close to the isotopic composition of soil organic matter that varies between -24 and -34‰ (Brunet et al., 2009). For the catchment with its runoff in dry seasons dominated by groundwater inputs, the more depleted $\delta^{13}$C in spring demonstrated

the contribution of $CO_2$ in soil water to $CO_2$ emissions. In comparison, the $\delta^{13}$C values were

comparatively enriched in summer and autumn (Table 2; Fig. 9), which probably suggests the impact of decomposition of C4 plants that have a $\delta^{13}C$ end-member of -12‰ (Brunet et al., 2009). Constrained by dry climate, major crops in the catchment are predominantly C4 plants, such as corn and millet, and their growing season from May until October overlaps well with the

summer and autumn samplings. Thus, decomposition of these $^{13}C$-enriched organic matter in summer and autumn resulted in more positive $^{13}C$ than that in spring. In addition, $CO_2$ diffusion process itself can induce isotopic fractionation (Deirmendjian and Abril, 2018;Geldern et al., 2015). Preferential outgassing of $^{12}CO_2$ may have also contributed to the more depleted $\delta^{13}C$ values in the emitted $CO_2$ than that of the C4 plants. Aquatic algae with their $\delta^{13}C$ value ranging

from -40‰ to -26‰ (Alin et al., 2008) is likely another contributor, as suggested by the 2-fold higher Chl *a* contents in summer and autumn than in spring at some sites (Ran et al., 2017). Deeply incised stream channels provide favorable stagnant water, albeit highly site-specific, for algae growth during non-flooding periods. However, this process seems to be of minor importance given the low light penetration due to extremely high turbidity.


As a useful tracer, natural radiocarbon has been widely used in terrestrial, aquatic, and marine carbon studies to trace the nature (i.e., age and source) and processing of carbon during transit (Gillon et al., 2012;Hemingway et al., 2017). The $^{14}C$ exhibited a weak negative correlation with $\delta^{13}C$ and the $^{14}C$ age increased from autumn through summer to spring (Fig. 9). Because DIC

from carbonate dissolution is characterized by typically enriched $\delta^{13}C$ and highly depleted $^{14}C$ (Mayorga et al., 2005;Brunet et al., 2009), distribution of the sampled $CO_2$ in this dual-isotope plot also suggests the negligible contribution of carbonate dissolution to $CO_2$ emissions. Instead, in spring dominated by groundwater influx, aged soil-respired $CO_2$ and decomposition of old OC leached from deep soil horizons have likely led to the older $^{14}C$ age (Figs. 5 and 9). This suggests

that the emitted $CO_2$ is derived from ancient terrestrial OC which is mineralized either in soils and then transported into rivers or in aquatic systems during transit (McCallister and del Giorgio, 2012). Addition of recently-fixed organic matter in summer and autumn through surface water inputs and decomposition of the bioavailable fraction have likely played a 'dilution' effect, causing the younger $^{14}C$ age and thus the seasonal distinctions. Notably, the emitted $CO_2$ is

inconsistent with that from the tropical Amazon rivers where respiration of contemporary organic matter is the primary source of excess $CO_2$ (Abril et al., 2014;Mayorga et al., 2005). Therefore, special efforts are needed to quantify this old $CO_2$ outgassing and assess its significance for global carbon cycle and climate mitigation over longer timescales than recent sharp anthropogenic $CO_2$ emissions (i.e., since the 1850s).


### 4.4 Riverine carbon budget and NEP

Our first-order estimate of NEP for the Wuding River catchment indicates that its terrestrial ecosystems sequester only small quantities of carbon. Approximately 83% of the NPP was consumed by microbial activities. This ratio is comparable to the estimate for global temperate

semiarid ecosystems (i.e., 84% from Luyssaert et al., 2007) while significantly higher than that for other ecosystems. For example, it is 63% in the tropical Nyong River catchment in western Africa (Brunet et al., 2009) and 42% in the temperate Schwabach River catchment in Germany (Lee et al., 2017). The total carbon into the Wuding river network is $(18.5\pm4.5)\times10^{10}$ g C year$^{-1}$, amounting to 16% of its catchment NEP (Fig. 10). This percentage of NEP as fluvial export is

also substantially higher than recent studies in other regions which found that the sum of DOC, DIC, and $CO_2$ emissions generally represented <3% of the NEP (e.g., Brunet et al., 2009; Lee et al., 2017). Although POC flux and OC burial are not quantified in these studies, the missing amounts are small due to weak soil erosion and absence of dams in their catchments. Similarly, Shibata et al. (2005) found that the annual export of dissolved and particulate carbon from a first-

order catchment in northern Japan made up only 2% of its NEP. However, the estimated NEP in this study is likely associated with large uncertainty as shown in Fig. 10. While a ratio of 40% of $S_R$ was used to calculate $R_h$ in non-forested areas, it could vary from 10% to 90% depending on land cover type (Hanson et al., 2000). For example, if the ratio is reduced to 35%, the proportion of total lateral export to NEP would decrease by 5.6%. Further research involving field

experiments and remote sensing technique is needed to constrain this estimate.

These discrepancies between Wuding and these catchments likely reveal the internal differences in soil property and erosion. Erosion-induced mobilization of heavily weathered soils with high calcite content into the Wuding river network exhibit a high DIC concentration and percentage

flux (Fig. 10). Compared with these catchments with weak soil erosion, the strong soil erosion intensity in the Wuding River catchment mobilized huge quantities of carbon into the river network. OC burial through sediment storage plays a significant role in re-distributing the exported carbon (Fig. 10). Shibata et al. (2005) did not quantify $CO_2$ emissions, which can be exceptionally higher than lateral fluxes, especially in first-order streams with strong boundary

turbulence (Marx et al., 2017).

While the proportion of total fluvial export to NEP in this catchment (i.e., 16%) is higher than other catchment-based estimates, it is substantially lower than the global-scale estimate of 50–70% by Cole et al. (2007). Compared with other ecosystems, the arid-semiarid Wuding River

catchment has a lower terrestrial NEP but a higher carbon export rate because of severe soil erosion. The resulting 16% likely represents the upper limit of the proportion of fluvial carbon export to terrestrial NEP. Thus, the conservative estimate by Cole et al. (2007) may have overestimated the importance of fluvial export in modulating terrestrial carbon uptake (Lee et al., 2017). Although 16% of the annual NEP was exported into the Wuding river network,

approximately 42% of it was buried behind check dams and sequestered thereafter. Given the rapid sedimentation and subsequent land management (i.e., cropland reclamation), this OC burial could be regarded as a long-term carbon sink (Zhang et al., 2016;Wang et al., 2011;Wang et al., 2017). Carbon loss through $CO_2$ outgassing can offset only 3% of the catchment NEP (Fig. 10). However, this first-order calculation may have underestimated carbon loss because the exported

carbon exiting the river mouth is subject to further processing and emission.

From a mass balance point of view, our analysis shows that more carbon was buried in sediments than was emitted as $CO_2$ from rivers and check dam-formed reservoirs in the Wuding River catchment. The 2-fold higher OC burial than $CO_2$ emissions is partially due to the strong soil

erosion and high sediment trapping efficiency of check dams, resulting in high OC burial rates (Mendonça et al., 2017). Another reason is the low drainage density of the river network governed by dry climate, leading to a small extent of water-air interface for $CO_2$ emissions, though the areal $CO_2$ emission fluxes are similar in magnitude to rivers in other climate zones

(Ran et al., 2017;Wallin et al., 2013). However, this comparison was based only on $CO_2$ emissions, since $CH_4$ emissions were not accounted for in the budget, although its contribution is likely negligible owing to high sedimentation rates, low water temperature, and low OC content.

**5. Conclusion**

The Wuding River catchment serves as a unique arid-semiarid study area for assessing the fate of terrestrially derived riverine carbon. Export of riverine carbon was predominantly composed of DIC due to widespread carbonate dissolution and groundwater input. DOC export was characterized by spatial variability. Continuous mineralization of the bioavailable fraction of DOC has probably caused the spatially downstream decline in DOC concentration in low order streams. In addition, the predominance of groundwater input over the entire year may has likely explained the seasonal insensitivity of DOC concentration to flow dynamics. POC% displayed strong seasonal variability throughout the catchment or at the catchment outlet, indicating the control of gully erosion in wet seasons in mobilizing deeper soils with low carbon content. The POC flux is comparable to the DIC flux on an annual basis, both of which are an order of magnitude larger than the DOC flux.

$CO_2$ emissions are quantitatively important, amounting to 20% of the total riverine carbon flux. Carbon isotopic analysis showed that the age of the emitted $CO_2$ ranged from 810 to 1890 years. Outgassing of this old carbon previously stored in soils has important biogeochemical implications for carbon budget studies. Our first-order estimate suggests that the riverine carbon export from terrestrial ecosystems was significant when compared with NEP, representing 16% of the latter. Riverine carbon cycle in the Wuding River catchment has been greatly modified by check dams through sediment storage. Approximately 42% of the total riverine carbon was buried, roughly twice the carbon loss through $CO_2$ emissions. With more new check dams under construction, OC burial will be a more vital component in reshaping the carbon balance. In addition, episodic storms play a disproportionate role in annual carbon export and future sampling strategy should attempt to capture these short-duration, high-discharge events to better constrain uncertainty.

Through a comprehensive assessment of riverine carbon in terms of downstream export, OC burial in sediments, and $CO_2$ emissions in a complete catchment, the present research can be treated as an exploratory study integrating river carbon cycle with terrestrial carbon uptake by ecosystems. A better understanding of linkages between terrestrial ecosystems and fluvial carbon export, and of interactions between environmental controls and human impacts, is essential for providing additional constraints on the accuracy of carbon budget estimates. Moreover, for future studies of riverine $CO_2$ emissions, it is critical to trace its isotopic composition and age to more holistically explore its biogeochemical significance.

**Acknowledgements:** This work was supported by the University of Hong Kong (grant: 104004330), the Natural Science Foundation of China (grants: 91547110 and 41671282), and the National University of Singapore (grant: R-109-000-191-646). The data used are available in the Supplement and Ran et al. (2017). Special thanks go to Jordon Hemingway, Minjin Lee, and an anonymous reviewer for their constructive comments which greatly improved the manuscript.

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

Table 1. $CO_2$ emissions from 337 check dam-formed reservoirs within the Wuding River catchment (±1SD).

| Subcatchment | Spring | Summer | Autumn | Spring (120 d) | Summer (123 d) | Autumn (122 d) |
|---|---|---|---|---|---|---|
| | mmol m$^{-2}$ d$^{-1}$ | | | million mol $CO_2$ year$^{-1}$ | | |
| Sandy subcatchment | 28.0±36.2 | -12.0±19.3 | 15.3±5.6 | 107±138 | -47±75 | 59±22 |
| Loess subcatchment | -2.9±9.9 | -17.4±14.8 | 11.5±17.6 | -26±89 | -161±137 | 106±161 |
| Total | | | | 81±165 | -208±156 | 165±163 |

Table 2. Carbon isotope signature of the emitted $CO_2$ from the Wuding River catchment (±1SD).

| Site | Spring | | | Summer | | | Autumn | | |
|---|---|---|---|---|---|---|---|---|---|
| | pMC | Age ($^{14}$C year BP) | $\delta^{13}$C (‰, VPDB) | pMC | Age ($^{14}$C year BP) | $\delta^{13}$C (‰, VPDB) | pMC | Age ($^{14}$C year BP) | $\delta^{13}$C (‰, VPDB) |
| S1 | 82.3±0.3 | 1560 | -32.3 | 88.0±0.3 | 1030 | -33.9 | 84.2±0.3 | 1380 | -24.4 |
| S2 | 79.0±0.3 | 1890 | -27.5 | 84.0±0.3 | 1400 | -22.2 | 86.0±0.3 | 1220 | -19.9 |
| S3 | 85.1±0.3 | 1290 | -26.5 | 90.4±0.3 | 810 | -22.7 | 90.3±0.3 | 820 | -25.2 |
| S4* | 80.9±0.3 | 1700 | -34.3 | 89.3±0.3 | 910 | -19.3 | | | |

*Sample for site S4 in October was lost during treatment.

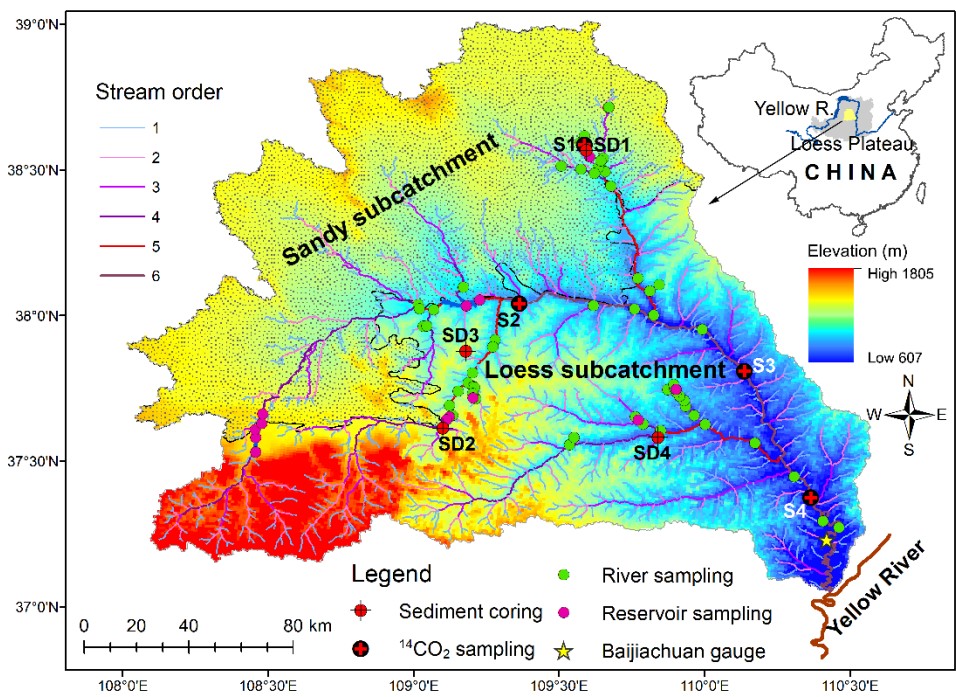

Figure 1. Map of the Wuding River catchment showing the sampling sites. SD1−SD4 and S1−S4 denote the sampling location of sediment coring behind check dams and carbon isotope, respectively. The inserted map shows its location on the Loess Plateau.


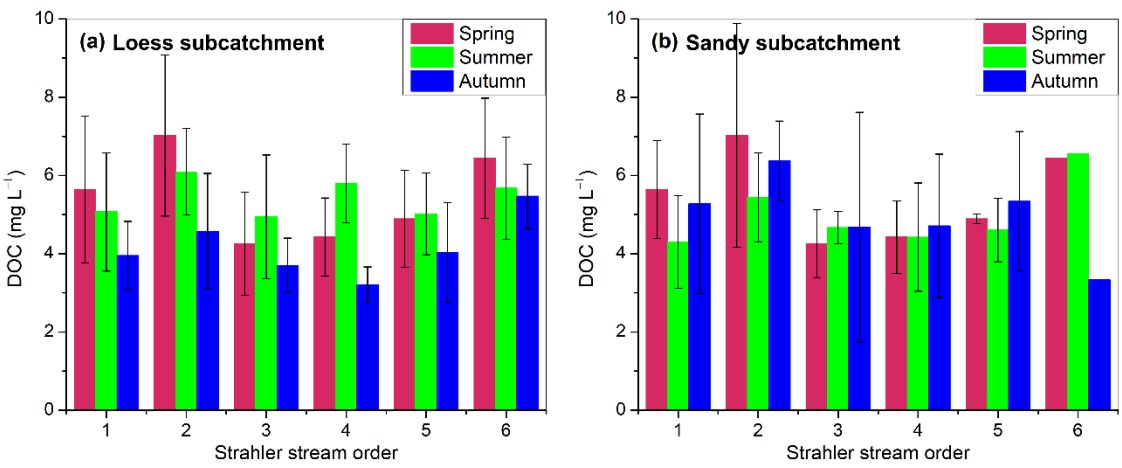

Figure 2. Spatial changes in DOC along the 6 Strahler stream orders in (a) the loess subcatchment and (b) the sandy subcatchment. Error bars denote the standard deviation (±1SD).

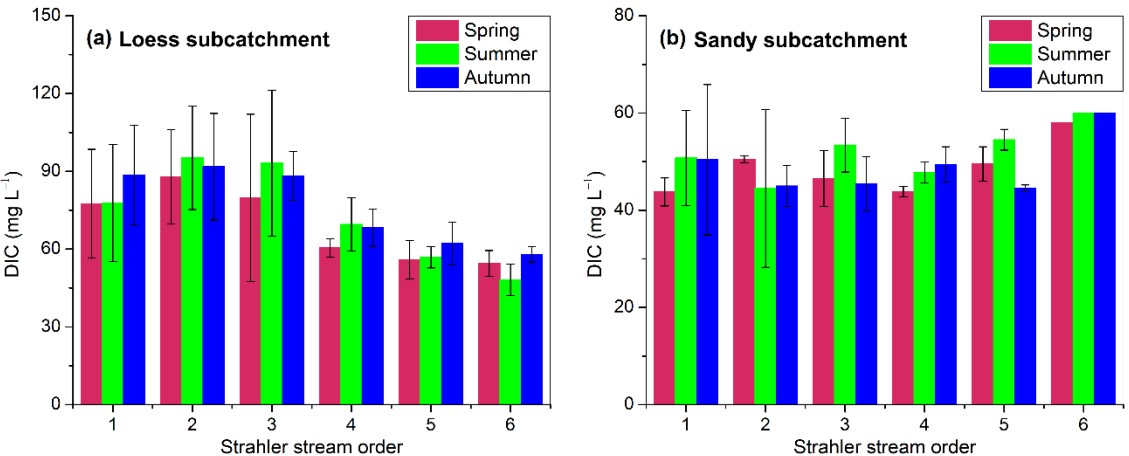

Figure 3. Spatial changes in DIC along the 6 Strahler order streams in (a) the loess subcatchment and (b) the sandy subcatchment. Error bars denote the standard deviation ($\pm$1SD).

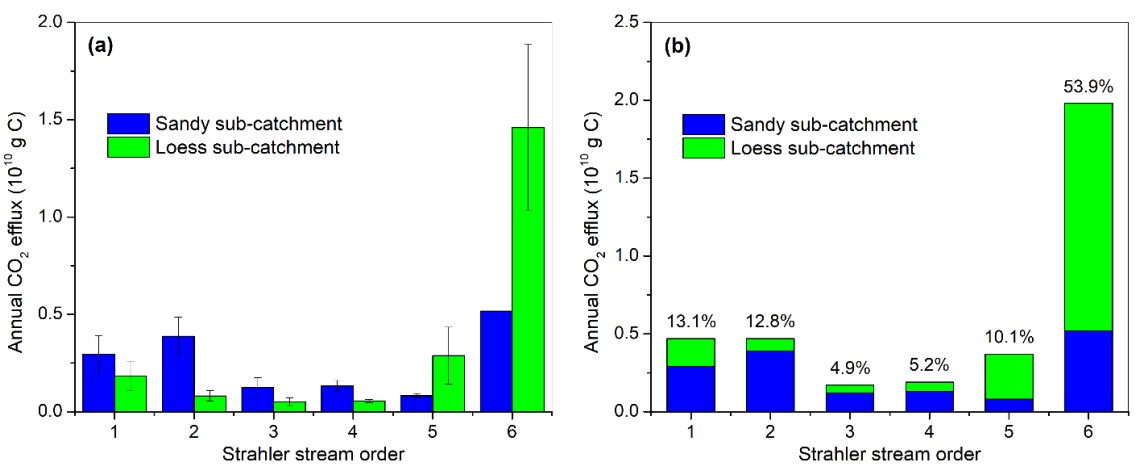

Figure 4. Longitudinal changes in $CO_2$ emissions along stream order in (a) the sandy subcatchment and the loess subcatchment and (b) the entire Wuding River catchment. The percentage above each order in (b) represents the proportion of $CO_2$ emissions from that order streams to the total $CO_2$ emissions. Error bars denote the standard deviation ($\pm$1SD).

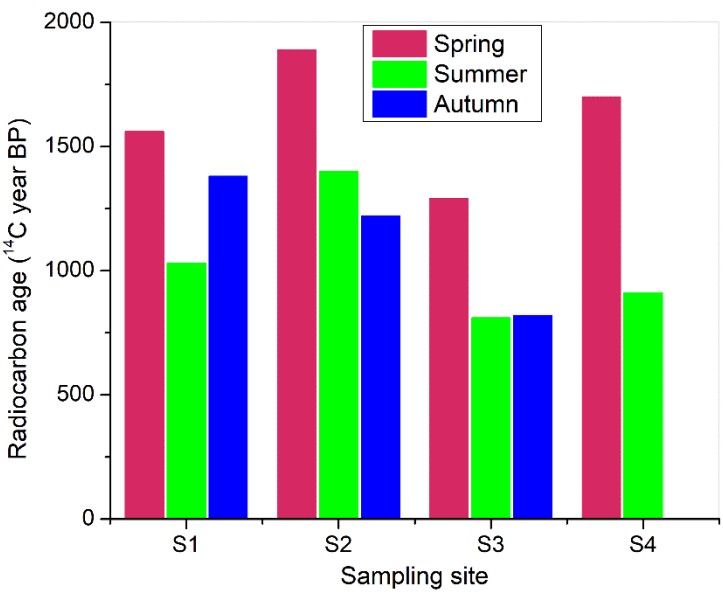

Figure 5. Seasonal variations in radiocarbon age ($^{14}$C year BP) of the emitted $CO_2$ from the
Wuding River catchment.

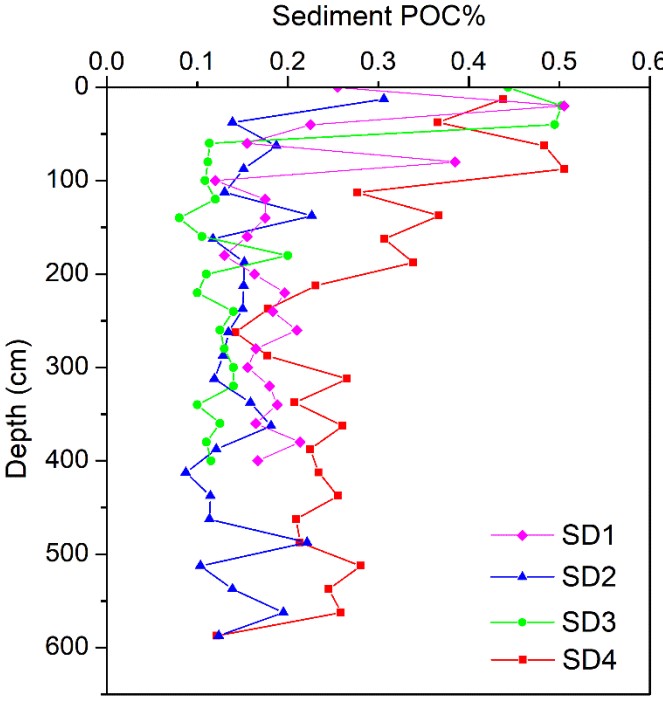

Figure 6. Variations of POC% with depth in buried sediments behind check dams (refer to Figure
1 for their location).

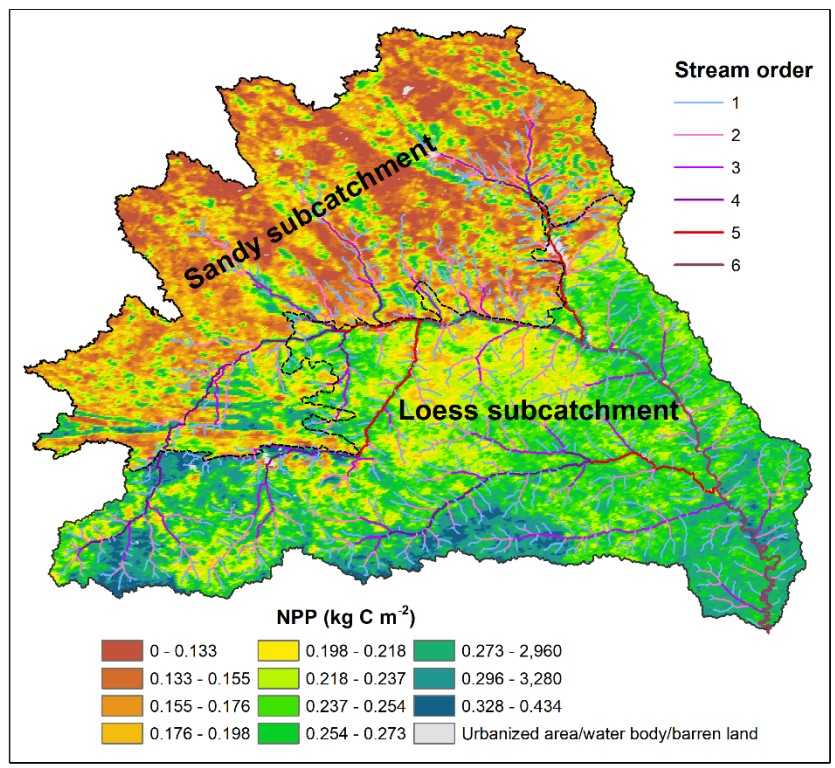

Figure 7. Spatial distribution of NPP within the Wuding River catchment in 2015 showing
significant differences between the northwestern sandy and southeastern loess subcatchments.

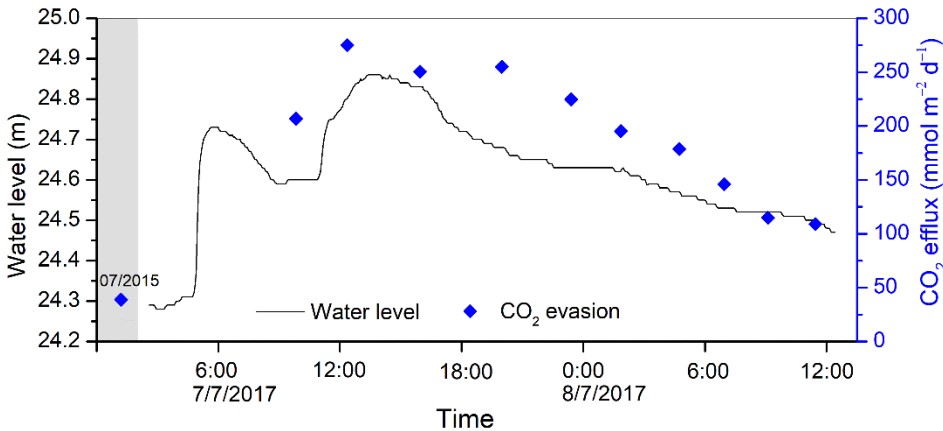

Figure 8. Temporal variation in $CO_2$ efflux during a high-discharge flood event in the Wuding
River at Baijiachuan gauge (refer to Figure 1 for its location).

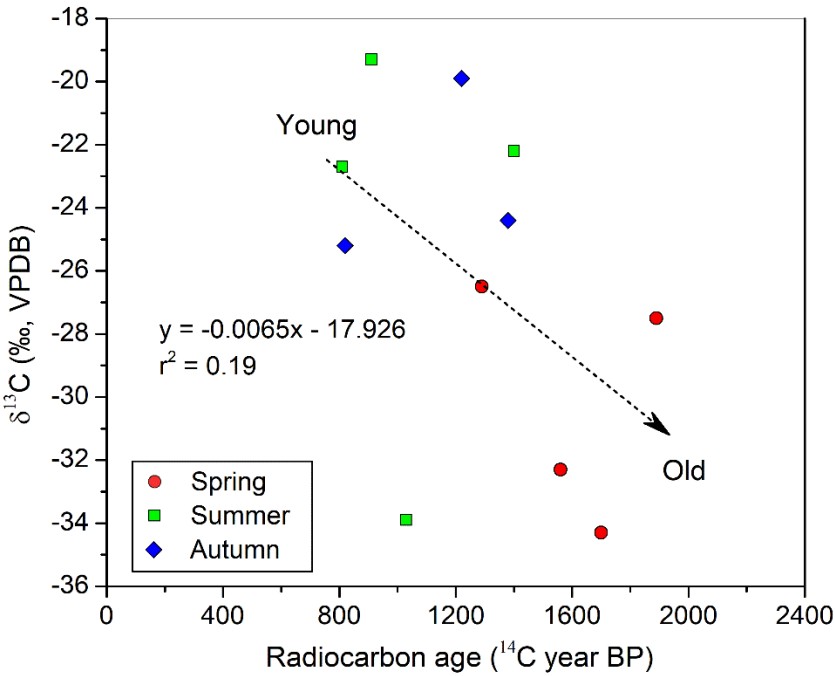

Figure 9. Relationship between $\delta^{13}C$ and radiocarbon age of the emitted $CO_2$ from the Wuding River catchment.

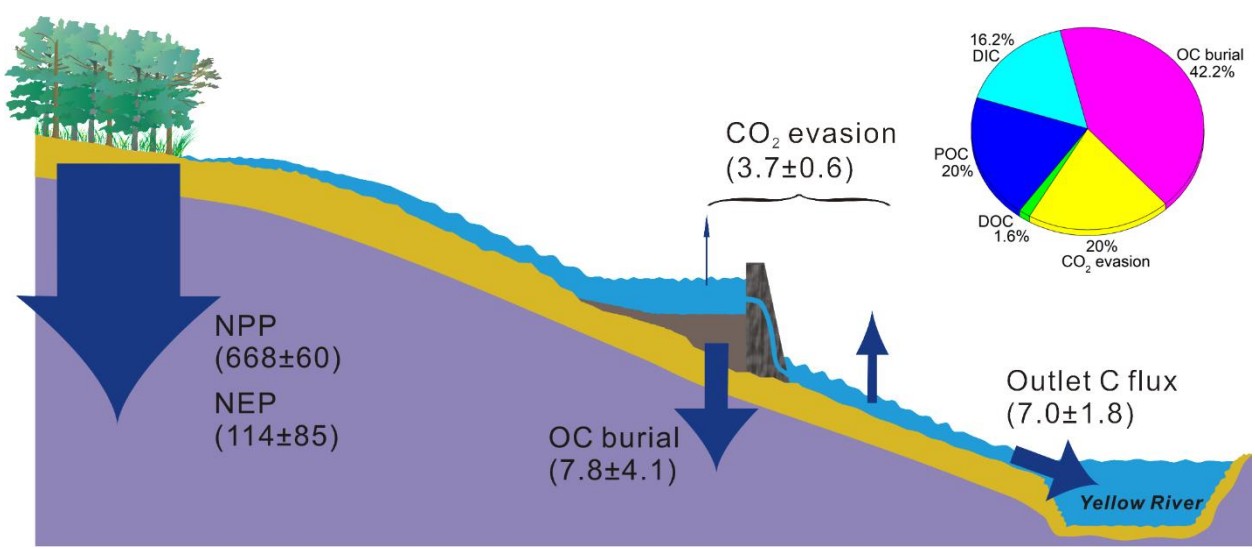

Figure 10. Fluvial carbon budget within the Wuding River catchment in relation to terrestrial ecosystem production (unit: $\times 10^{10}$ g C yr$^{-1}$). The inserted pie chart denotes the partitioning of riverine carbon among its five phases with the sum (100%) representing all the carbon entering the river network.