# Peer review of "Riverine carbon export in the arid-semiarid Wuding River catchment on the Chinese Loess Plateau"

_Biogeosciences, 2018_

## Referee Comment (RC1) · Anonymous Referee #1 · 5 Mar 2018

General comments:

Ran et al. reported new data on riverine carbon export in the arid-semiarid Wuding River watershed on the Chinese Loess Plateau. Considering that river systems in the East Asia, especially those in the arid-semiarid climates are under-represented in the global budget of riverine carbon fluxes, this study could provide valuable datasets. However, the paper can be improved further by explaining in detail how the errors were calculated in load estimates and CO2 evasion, offering detailed explanation in the methods (e.g. the river surface area), and providing discussion on the observed patterns with statistical significance testing results. Specific comments are below, which the authors may consider when revising the manuscript.

Specific comments:

Lines 48-: "substantial" is a relative term. Please provide a value or a range just like 1.8 Pg C year-1 in the previous sentence.

Lines 84-: "multi-annual" is an unspecific term. Please provide more information on how the mean of water discharge was calculated. For example, you can provide the period (e.g. 1980-2010?)

Also, year-1 as a time unit would be appropriate for annual discharge. Is it 11.2 * 10^8 m3 yr-1 (Ran et al., 2017)?

Lines 149-: Isn't the 14C half-life 5,730 years?

Lines 156-158: Detailed explanation is needed on the validity of the methods on how the riverine carbon exports were calculated considering that the major findings of this paper are the new estimates of the riverine carbon loads. Detailed explanation is needed on how river flow was measured. The method of load estimation appears to be too simple and with many assumptions, not specifying errors associated with each step. There are several methods for load estimation (e.g. Sickman, J.O. et al., 2007, Water Resources Research, Effects of urbanization on ...) you may try these and compare the results because load calculation is crucial to draw conclusions. One way to calculate daily load of stream ions is to use the LOADEST software developed by USGS if daily water discharge data are available. The software also provides confidence intervals.

Lines 160-: Do you mean the POC concentration not "content"? It appears the term, "content" is misused throughout the manuscript.

Lines 170-: How large is the river width? If it is near or lower than 90 meters, how can you estimate river surface areas using the DEM data of 90-m resolution? In other words, aren't you using too coarse data to estimate river water surface areas?

More detailed explanation is required on how the water surface area is calculated since this is a critical term for CO2 evasion estimates.

Lines 183-198: The method is better than nothing for sure. However, it appears the used references are relatively old (1995 and 2000). Do you have newer references on heterotrophic respiration than those? How the errors associated with the approach are calculated?

Lines 208-: Does the 'sediment' mean 'suspended sediment'? If so, please clarify it to prevent confusion.

Do you mean the POC concentration not "content"?

Lines 222- and throughout the manuscript: What is the "+/-"? Standard deviation? Or standard error?

Lines 225-: While [DOC] (3.3 mg/L) is larger than [POC] (0.61 mg/L), the DOC export is much lower ($0.3*10^{10}$ g (yr-1?)) than POC export ($3.7*10^{10}$ g (yr-1?)). Why is that so?

Lines 228-233: The river water discharge and carbon loads can be highly dependent on precipitation. Was the year of field campaign categorized as wet, dry, or normal year compared to the long term mean (e.g. 1980-2017 precipitation)?

Lines 258–261: As the authors mentioned, the precipitation is high during summer. Thus, this assumption of no significant seasonal fluctuations may not be valid. Can you provide a range of stream surface area and $CO_2$ evasion depending on season?

Lines 300–: Is the decreasing trend of DOC (Fig. 2) statistically significant? It appears the error bars are large. If this is not statistically significant, the following argument is vague. The decrease of DOC can be microbial- or photo-degradation to $CO_2$, sorption to particulate matter, and dilution from increased water discharge of low [DOC]. The following discussion is speculative and could be strengthened by checking each factor.

Line 325–336 (and lines 385–394, and Fig. 6): I am confused. Do you mean the POC concentration not "content"? Why the "content" has the unit of concentration, %, not just grams? I think heavy rain during summer could generate high POC content but

low POC concentration. Please clarify.

Line 351–368: The pCO2 is a function of pH and alkalinity. The pCO2 is high when the water pH is low. The ground water of the area has the pH of $>\sim8$. Then, the calculated pCO2 is very low which is well described in the line 211. Then, how CO2 evasion can be high when pCO2 is low? Please clarify.

Lines 400: Which part of the Figure S1 supports this sentence?

Lines 430-481: Very interesting findings.

Tables: : What is the "+/-"? Standard deviation? Or standard error?

Table 1: Please provide information on how many reservoirs were used to draw the table.

Figure captions need to provide more detailed description of the figures including explanation on legends.

Figure 1: It is hard to differentiate the colors of the stream order, especially with the background altitude colors. Please revise the figure so that each symbol can be seen clearly.
* * *

---

## Referee Comment (RC2) · J. Hemingway (Referee) · 6 Mar 2018

*Review of Ran et al. "Riverine carbon export in the arid-semiarid Wuding River catchment on the Chinese Loess Plateau" (bg-2018-51)*

Synopsis
      The central focus of this manuscript is to investigate carbon cycling in the arid-semiarid Wuding River catchment using both campaign-style and time-series sampling approaches. The authors quantify dissolved carbon concentrations, both organic (DOC) and inorganic (DIC), as well as particulate organic carbon (POC) concentrations and $CO_2$ outgassing fluxes throughout the catchment over multiple seasons. In particular, the authors compare and contrast signals across a range of Strahler stream orders (1 to 6) from subcatchments underlain by sand and by loess and quantify differences in their respective carbon budgets.

      As the authors point out, arid-semiarid river catchments are severely underrepresented in global riverine carbon-cycle budgets. By presenting a large dataset for the Wuding River catchment, this study begins to ameliorate this issue. I therefore find the goals and targets of the present study to be impactful, as they attempt to advance our collective understanding riverine carbon cycling. However, I do have some issues with the interpretation of these data, particularly related to a number of claims that seem unsubstantiated or somewhat contradictory. Additionally, I feel that there are some areas that warrant further clarification and detail. Overall, I feel that the authors should remove some of the weaker and highly speculative text that attempts to prescribe carbon sources and should instead focus on the strengths of this dataset – namely, carbon fluxes and budgets. If the authors can address these issues, which I think they can, then I believe that this manuscript could provide a valuable contribution to *Biogeosciences*.

      I outline my larger concerns in detail below, followed by a list of smaller concerns and questions. Please do not hesitate to contact me for further discussion regarding this review.

Sincerely,

Jordon Hemingway

jordon_hemingway@fas.harvard.edu

**Larger Comments**

*Methods details and measurement uncertainty*
      In general, I feel that more detail is required in describing the methodology and presenting data uncertainty. In particular, the paragraph beginning on L135 should be expanded considerably. For example, I would like to see more details related to:

i) Field titration methods. How was this done? Were any standards measured? Field titrations generally have quite high uncertainty associated with them (~5 – 10%), yet there is no uncertainty assessment presented here. What is the resulting propagated uncertainty for calculated DIC concentration values?

ii) DOC uncertainty. How was DOC uncertainty estimated? Was a standard calibration curve used? If so, how often was the calibration curve analyzed? Was each sample injected in triplicate? Duplicate? Single injection?

*iii)* Were solid samples fumigated with HCl at room temperature or at ≥60°C? I ask because dolomite will not be removed at temperatures below 60°C. If these samples are expected to contain dolomite, and if they were fumigated at room temperature, then I would expect resulting POC estimates to be biased upward.

*iv)* $CO_2$ $\delta^{13}C$ values. Were these analyzed by Beta Analytic using an IRMS on a separate gas split, or are these values generated by the AMS? I would expect these to be IRMS values, but this should be stated clearly.

*v)* Radiocarbon notation. Throughout the manuscript, the authors conflate $^{14}C$ age, $\Delta^{14}C$ (which is *always* reported in units of per mille!) and percent modern, or pMC. I would strongly suggest that the authors choose one notation and stick with it (my personal choice would be to use pMC). Still, if the authors choose to use $^{14}C$ age, this be reported in units of "$^{14}C$ yr BP" rather than simply "years", as the latter is ambiguous and could refer to a calibrated age, which would not be appropriate here.

*vi)* Sediment accumulation rates. In Figure 10, a burial flux is presented in units of g C $yr^{-1}$, yet I find no reference to calculations for sediment accumulation rates (SAR). How was SAR calculated for each of these cores? This information is necessary in order to convert the measured %OC numbers into burial fluxes…

Additionally, all of the numbers reported in the "Results" section should include corresponding uncertainty, either analytical uncertainty (when reporting single values) or sample population uncertainty (when presenting averages). For averages, please be clear if reporting standard errors or standard deviations. Similarly, significant figures should be consistent throughout the manuscript!

*Net Ecosystem Production*

      I am left somewhat confused by the assumptions and uncertainties related to NEP calculations. To convert $S_R$ to $R_h$, the authors apply a "forested" and "non-forested" fraction heterotrophic derived from Hanson et al. (2000). However, the "non-forested" estimates from this reference are for pasture and grassland, not barren landscapes such as those presented in the current study. Presumably nearly 100% of soil respiration on barren landscapes is heterotrophic, no? Additionally, while the "non-forested" fraction heterotrophic in Hanson et al. averages 40%, they observe values ranging from 10% to 90% -- nearly the entire possible range!

      I wonder if the authors have any way to estimate the uncertainty on NEP estimates presented here – if so, these should be discussed in detail. I would expect these uncertainties to be quite large, yet this is not mentioned or discussed in the manuscript. For example, how do the values here compare to those calculated by subtracting $S_R$ from MODIS-derived GPP values? To me, this seems like a more straightforward method to estimate NEP that isn't subject to the uncertainties associated with converting $S_R$ to $R_h$.

*Interpretation of DIC, $CO_2$ $\delta^{13}C$, and $\Delta^{14}C$*

      In general, I am confused by the discussion on DIC sources, especially as they relate to measured $CO_2$ $\delta^{13}C$ and $\Delta^{14}C$ values – there seem to be a number claims that are either contradictory or are not explained in significant detail. Beginning in the abstract (L21) and repeated throughout the manuscript, the authors state that DIC is largely sourced from carbonate dissolution, especially in the loess subcatchment. Intuitively, this makes sense to me since loess contains a significant amount of carbonate, as the authors rightly state. However, this is

incompatible with the $\delta^{13}C$ and $\Delta^{14}C$ values presented in this study, which suggest that remineralization of terrestrially derived OC is the main source of outgassed $CO_2$ behind check dams. What mechanisms could explain this discrepancy? I feel that there needs to be significantly more discussion and clarification here.

Furthermore, I find some of the claims related to $CO_2$ outgassing to be overstated. For example, the statement: "The evasion of old carbon [derived from pre-aged OC respiration as is seen here] is likely to be widespread in arid-semiarid catchments worldwide with similar hydrological regime and terrestrial ecosystems" (L477). This seems to be quite a stretch, especially given my confusion related to the lack of carbonate dissolution signature as stated above.

*DOC sources and trends*

Beginning on L204 and continuing throughout the manuscript, the authors refer to a "significant downward trend along the river course from headwater downstream… in both subcatchments." However, when I look at Figure 2, I am left puzzled and wondering if these trends are, in fact, significant. Given the large error bars for each stream order, my guess is that they are not. In my opinion, any subsequent discussion related to DOC sources and trends (e.g. L300-313; L306-309; L318-324) is highly speculative at best.

Additionally, I find some of these claims to be contradictory. For example, on L314, the authors state that "…there was no significant correlation between DOC and flow based on the spatial sampling results". However, for the high-frequency sampling the authors observe a "positive correlation between DOC export and hydrography [that] demonstrates the enhanced leaching of organic matter from surface vegetation and organic-rich top soil layers". Why would a positive correlation be expected during storm events yet not on a seasonal basis? What mechanism could explain this? This discrepancy is not addressed.

*POC sources and trends*

I find that a significant amount of discussion related to POC sources and sinks needs to be substantiated with more evidence or, at a minimum, alternate explanations need to be addressed. First, beginning on L326, the authors claim that low POC content (by which they mean % of suspended solids, a point that I address below) "reflects the ancient sedimentary OC origin of about 0.5% for fluvial sediments worldwide… [and is also] seen from the isotopic signature of the Yellow River sediment…" The authors go on to state that low %OC reflects "mobilization of subsurface soils that have a substantially lower OC content than surface soils" (L334). However, "ancient sedimentary OC" presumably refers to sedimentary rock derived material, which is *certainly* not the same as "subsurface soils". I'm left confused as to what the authors expect to be the major source of POC – sedimentary rocks or subsurface soils? I think that, with concentration measurements alone, one cannot make strong claims either way.

The well-known relationship between grain size and %OC is also not addressed. The observed POC concentration trends could easily be explained by variable hydrologic sorting – i.e. coarser, OC-poor sediments that are transported during high discharge periods – which would mask any POC source signal. In the absence of isotopic ($\delta^{13}C$, $\Delta^{14}C$) or grain-size-dependent measurements (e.g. %OC as a function of Al/Si ratios), I find it hard to believe that POC sources can be prescribed as is done here (also repeated beginning on L385).

Similarly, beginning on L408, %OC content behind check dams is compared to that on hillslopes and is used as evidence for burial efficiency. However, this "negligible OC loss after

erosion" (L412) could be explained by alternative hypotheses. For example, deposited material could (likely does?) contain a different grain size distribution than that of hillslope soils, and thus a different %OC content. Also, any remineralization of terrestrially derived POC could be masked due to replacement by aquatic sources (as is discussed). Again, I find it hard to prescribe POC sources and burial efficiencies without additional measurements such as $\delta^{13}C$ and $\Delta^{14}C$. I also find the claim that this material "would have otherwise been mineralized to form $CO_2$ or $CH_4$ along fluvial delivery" (L418) to be somewhat speculative. Presumably some of this material would have been transported and buried in coastal marine sediments. Heuristically, it makes sense that burial efficiencies behind check dams are higher than for coastal marine sediments, as the authors imply, but I find a general lack of evidence supporting this claim.

Finally, I find that reporting "OC content" as %OC rather than a concentration (e.g. mg OC $L^{-1}$) or a flux (e.g. t OC $km^{-2}$ $d^{-1}$) is ineffective and is somewhat misleading. For example, the authors state that "the substantially lower POC content in the wet season largely reflects the impact of gully erosion" (L385). However, one would expect that POC *concentration* and *flux* are actually significantly higher during the wet season! As described above, changes in %OC could reflect hydrologic sorting and are not necessarily indicative of source. I would strongly recommend discussing POC trends in the context of concentration and flux, rather than %OC. This would allow the authors to shift the focus away from attempting to prescribe POC sources (which I find to be a weakness overall) and toward OC flux and budget estimates, which I think is a strength of this manuscript.

*Data availability*

In my opinion, a major strength of data-rich manuscripts such as this is the ability for readers to incorporate these data into future studies – whether those be review articles or comparisons to other, similar catchments. Along those lines, I am left wondering why the authors do not make all of their raw data available as supplemental tables? I would strongly suggest do so or, at a minimum, including a "Data Availability" statement pointing the reader to a repository that includes these data.

**Smaller Comments**

L14: Remove dash between "terrestrially derived", change "represent" to "represents".

L15 (also L68): What is meant by "redistribution"? Do the authors mean "partitioning between DIC, DOC, and POC"? I would change this wording for clarity.

LL17: Change to "While DOC…"

L18: What is meant by "DOC concentration is spatially comparable within the catchment"? I'm confused by this statement. Don't you argue that DOC concentrations decrease with increasing stream order? (although I question this trend, as stated above).

L19: "This reflects the enhanced…" seems overly confident. I would say "This *likely* reflects…"

L21-22: I'm still confused by the DIC sources – carbonate dissolution seems incompatible with the measured $CO_2$ $\delta^{13}C$ and $\Delta^{14}C$ values.

L23: Please be clear that you mean %OC in sediments when stating that "[POC content] shows low values in the wet season." As stated, this implies that POC concentration or flux are lower in the wet season, which I presume is not true.

L27 (and throughout): Please update the $^{14}$C notation, as described above. "Indicating the release of old carbon previously stored in soil horizons." Couldn't this also be described as a mixture of $^{14}$C-free carbonate dissolution and respired young OC? I'm not sure that this claim is supported.

L32: Define "NEP". I don't follow the last sentence of the abstract. What is meant by "…has been significantly offset by riverine carbon export"?

L38: "Rivers play an exceptionally significant role by directly linking…" Role in what? The global carbon cycle?

L39: Remove dash between "terrestrially" and "derived".

L43: add "the" between "along" and "river".

L44: Remove comma after "processes".

L45: Change "in-situ" to "*in situ*" for consistency.

L46: How up-to-date is this 1.8 Pg C yr$^{-1}$ number? See Drake et al. (2017) *L&O Letters* for an updated number.

L51: Has the number of studies on riverine carbon *really* been increasing *exponentially*? Change "recent" to "last".

L66: this should read "…through *the* drainage network to *the* catchment outlet…"

L67: Remove "in" before "northern".

L83: This should read "…and *is* located"

L85: Consider defining "loess" here.

L92: Citation for hydrologic regime description?

L94: I'm confused – is this sentence saying that one particular extreme event led to an erosion rate of 7000 t km$^{-2}$ yr$^{-1}$ for a particular year? If so, what is the average erosion rate? I feel like this would be more informative.

L102-103: "[The altered $CO_2$ exchange] remains to be quantified". Didn't you quantify this in Ran et al. (2017)? If so, how does this "remain to be quantified"?

L110: Change "was" to "is".

L124: Change "triple" to "triplicate".

L127: "radiocarbon $\Delta^{14}C$ samples" is somewhat redundant. Change to "collected samples for $^{14}C$ analysis" or similar.

L147 (and throughout): "The $\Delta^{14}C$ values were reported as percent modern carbon (pMC)". These are two separate units! (see above discussion).

L156: How reliable is this method for calculating carbon loads? This seems too simple. Why was something like LoadEst not used?

L160: "by multiplying annual sediment deposition rate…" How was deposition rate calculated? This is not described at all in the text.

L162: Change "was" to "were". Were these $CO_2$ flux data taken directly from Ran et al. (2017), or are these new data originally presented in this study? Overall, I would clearly state which data are new and which data are taken from previous studies (as these authors appear to have published multiple papers on this dataset…)

L172: Please provide the minimum catchment threshold area, as this will affect calculated Strahler stream order.

L177: How valid is the assumption that "each round of field sampling [is] representative of $CO_2$ emissions" for these four-month periods? What about for DIC, DOC, and POC concentrations – presumably you assume these are representative too?

L189: Heterotrophic soil respiration need not be due to bacteria – this could also be fungal or archaeal respiration. I would simply stick with "heterotrophs".

L216: Is this decline (insofar as it is statistically significant) really "remarkable"?

L225: I'm confused by the sentence beginning with "Because the flow regime in 2017 was significantly biased…" What is this saying? You applied the 2015 hydrological regime to the 2017 data?

L230: Fluxes should be in units of "g C $yr^{-1}$" (I'm assuming the "$yr^{-1}$" got dropped by accident).

L257: This should read "Assuming the water surface *area* remained constant…"

L265 (and throughout) Please add "VPDB" after "‰" when reporting $\delta^{13}C$ values.

L277: Should "soils" instead read "sediments"? How can sediment cores contain "soils"?

L300: How turbid are these rivers? If they are quite turbid, then I would expect that photochemical degradation is probably insignificant.

L301 (and throughout): Please change "labile" to "bioavailable" as this language is more consistent with our current understanding of OC decay dynamics.

L312: I'm confused by the statement "…and the mixture of carbon export from the two subcatchments." I thought the "6$^{th}$ mainstem channels" *are* the two subcatchments, which combine to form the 7$^{th}$ order Wuding River? Or have I misinterpreted this? (It is hard to see on Figure 1).

L330: Similarly, please change "biogeochemically refractory" to "persistent" in order to be consistent with our current understanding of OC decay dynamics.

L358 & 362: Phytoplankton are not aquatic plants. Please clarify this language.

L426: I'm confused by the inclusion of this sentence – what does it add to the discussion?

L464: Is the correlation between $\delta^{13}C$ and $\Delta^{14}C$ statistically significant? Figure 9 does not report the regression slope equation nor any statistics, so I have no way of gauging the strength of this relationship (I'm not even sure if the line drawn in Figure 9 is a regression line…) Please clarify.

L470: "…which suggests the outgassing of ancient terrestrial OC after entering aquatic systems". I'm confused here – OC itself cannot be outgassed. Does this refer to $CO_2$ generated from remineralization of old OC? If so, how do the authors know that this was remineralized *after entering* the aquatic system and not simply remineralized in soils and transported with soil pore waters?

L472: This claim (and others throughout the manuscript) isn't necessarily supported – I would urge caution when making concrete statements such as this. Rather, I would phrase this along the lines of "These results are consistent with…"

L510: I'm confused by the statement "…this percentage (16%) falls into the range of global-scale estimates of 50-70%..." 16% is *not* in the range of 50-70%... am I missing something?

L514: In what way is the estimate of Cole et al. (2007) "conservative"?

L516-517 (and 531): Please remove "… it is worth noting that".

L537: Remove the dash between "terrestrially" and "derived".

L548: "$CO_2$ emissions represented an important pathway…" An important pathway for what? Carbon loss from the landscape? I would change this to "$CO_2$ emissions are quantitatively important…" or similar.

Figure 1: This figure is hard to read given the current color scheme and marker sizes. I would consider changing the color scheme for clarity and making the markers significantly larger. Also,

please provide a catchment outline for the "sandy" and "loess" subcatchments, as this delineation is currently unclear.

Figures 2-3: I would consider writing "Loess Subcatchment" and "Sandy Subcatchment" above panels (a) and (b) so that the reader does not have to dig through the caption to understand what is presented.

Figure 4: I'm confused by what the percentage numbers represent. This should be clarified in the figure caption.

Figure 5: Why has the nomenclature and color scheme changed for this figure? Why not use the colored bars from Figures 2-3 and the "spring", "summer", "autumn" notation that is used throughout the text? Also, what is meant by "conventional age"? Is this equivalent to $^{14}C$ yr BP?

Figure 7: Why is this figure showing NPP when the authors are interested in NEP? Why not show NEP directly? Also, please include the river network, subcatchment outlines, labels, etc. as in Figure 1.

Figure 9: Why are units of pMC (which is not the same as $\Delta^{14}C$!) used in this Figure but $^{14}C$ years used in Figure 5? Is this dashed line a regression line? If so, please include the regression equation and statistics. Technically, "young" and "old" only correspond to the $y$-axis and should point vertically, as the $x$-axis of this figure says nothing about age.

Figure 10: I'm confused by the inset pie chart – what does 100% represent? Is this all of the carbon in the river network? If so, at what time points, or does this represent the relative annual fluxes? Again, more detail in the caption would be very much appreciated.

Figure captions: In general, I would like to see significantly more description in this figure captions.

---

## Short Comment (SC1) · 18 Mar 2018

This study provides rich river carbon data from a watershed influenced by arid-semiarid climate. The data, including river carbon concentrations, exports, contents, and emissions in different carbon species, are very informative. I believe that more careful analyses of these comprehensive data can enhance our understanding of river carbon cycling and its role in linking terrestrial and marine biogeochemistry. I found some small and large problems which I think should be addressed for publication of this manuscript in Biogeosciences.

Estimation method of river carbon exports P4L156-160: River carbon exports are one of key results of this study, and thus should be estimated very carefully. However,

[Figure]

I found that the estimate method of the exports is not clear. There are various estimation methods that could be applied. Aulenbach, B.T., Buxton, H.T., Battaglin, W.A., and Coupe, R.H., 2007, Streamflow and nutrient fluxes of the Mississippi-Atchafalaya River Basin and subbasins for the period of record through 2005: U.S. Geological Survey Open-File Report 2007-1080, https://toxics.usgs.gov/pubs/of-2007-1080/index.html Cohn, T.A., Caulder, D.L., Gilroy, E.J., Zynjuk, L.D., Summers, R.M., 1992, The validity of a simple statistical model for estimating fluvial constituent loads—An empirical study involving nutrient loads entering Chesapeake Bay: Water Resources Research, v. 28, no. 9, p. 2353–2363. Runkel, R.L., Crawford, C.G., and Cohn, T.A., 2004, Load estimator (LOADEST)—A FORTRAN program for estimating constituent loads in streams and rivers: U.S. Geological Survey Techniques and Methods, book 4, chap. A5, 69 p.

Estimation method and uncertainty of NEP P5L182-199 and P7281-291: For river carbon budget analysis, the NEP result is critical to drive the conclusion. However, I am a bit skeptical about the approach to calculate NEP. The authors are using different independent data sources for NPP and SR, and then, to calculate Rh, adapting another study's assumption "Rh accounts for 54% and 40% of SR in forested and non-forested areas,". This methodology probably led to large uncertainty in the final NEP estimate, which should be at least discussed.

Data availability and clarification A strength of this study is that it provides and interpret the very comprehensive river carbon data. Biogeosciences readers would be interested to see the data/results in more detail. There are many results which are described in texts, yet cannot be directly read by figures or tables. Also, the authors might want to have a simple table that lists the data with time (which year, season,...), units (concentration, contents, exports...), and brief estimation methods. This study covers a lot of interesting data, but I am confused by how they were presented. Also, I am confused by the use of "concentrations" and "contents".

P1L15: What do you mean by "redistribution"?

P1L17-18: I am not sure what you meant with this "While the DOC concentration was spatially comparable within the catchment," I would remove this.

P1L18-19 vs. P8L312-314: Is this sentence consistent with your claims in P8L312-314? I am confused. "it was generally higher in spring and summer than in autumn, especially in the loess subcatchment." vs. "There was no discernible seasonal difference in DOC concentrations in both subcatchments, although the hydrograph varied significantly among the three seasons."

P1L19-21 vs P8314-321 vs P9L375-377: I am also confused that these discussions appear to contradict each other. High soil carbon leaching due to high rainfalls in many cases leads to high river carbon exports (massC/time), but not high river carbon concentrations (massC/volumeH2O). High rainfalls increase river flows as well, so concentrations can increase or decrease.

P1L23 and P5L209: Did you mean "showed" by "shown"?

P2L84, P2L89, P2L94: An exact time period or years should be provided.

P6L225-228: The assumption should be justified better. Why did you particularly use hydrological data for 2015?

P7L298: Did you mean "concentrations" by "contents"?

P7L299: Specify by providing values to support "both DOC and POC contents in the Wuding catchment were relatively low compared with most rivers in the world."

P8L303: I am not sure if this statement is valid. "This decomposition is generally associated with increasing water residence time for bacterial respiration in downstream streams due to decreasing flow velocities." I don't think that flow velocity generally decreases toward downstream. I think that travel time generally increases toward downstream and longer travel times provide more opportunity for decomposition.

P8L326-328: I don't understand what you mean here.

---

## Author Comment (AC1) · 16 Apr 2018

Dear reviewer,

Many thanks for your comments on our manuscript. Based on your very constructive comments, we have thoroughly revised the manuscript. Additional discussion and justifications have been added into the manuscript or into the Supplement. Please see below the detailed responses. Major changes have also been highlighted in the revised manuscript.

With best regards
Lishan Ran, on behalf of the coauthors
* * *
General comments:
Ran et al. reported new data on riverine carbon export in the arid-semiarid Wuding River watershed on the Chinese Loess Plateau. Considering that river systems in the East Asia, especially those in the arid-semiarid climates are under-represented in the global budget of riverine carbon fluxes, this study could provide valuable datasets. However, the paper can be improved further by explaining in detail how the errors were calculated in load estimates and CO2 evasion, offering detailed explanation in the methods (e.g. the river surface area), and providing discussion on the observed patterns with statistical significance testing results. Specific comments are below, which the authors may consider when revising the manuscript.
Reply: Thanks a lot for your constructive comments. Please find below our responses to your comments.

Specific comments:
Lines 48-: "substantial" is a relative term. Please provide a value or a range just like 1.8 Pg C year-1 in the previous sentence.
Reply: The annual OC burial due to sediment storage in global reservoirs and lakes ranges from 0.15 to 0.6 Pg C year$^{-1}$. This range has been added into the revised manuscript. (lines 47-48)

Lines 84-: "multi-annual" is an unspecific term. Please provide more information on how the mean of water discharge was calculated. For example, you can provide the period (e.g. 1980-2010?) Also, year-1 as a time unit would be appropriate for annual discharge. Is it 11.2 * 10^8 m3 yr-1 (Ran et al., 2017)?
Reply: The calculated mean water discharge of 35 m$^3$ s$^{-1}$ is based on the period 1956–2017. The mean annual water discharge during this period is 11.2×10$^8$ m$^3$ year$^{-1}$. These changes have been added into the text. (lines 84-85)

Lines 149-: Isn't the 14C half-life 5,730 years?
Reply: The half-life used in carbon dating calculations by the Beta Analytic Inc. is 5568 years, the value worked out by chemist Willard Libby, and not the more accurate value of 5730 years, which is known as the Cambridge half-life. Although it is less accurate, the Libby half-life was retained to avoid inconsistencies or errors when comparing carbon-14 test results that were produced before and after the Cambridge half-life was derived. Detailed description on the 14C half-life can be found at the Beta Analytic website: https://www.radiocarbon.com/PDF/AMS-Methodology.pdf.

Lines 156-158: Detailed explanation is needed on the validity of the methods on how the riverine carbon exports were calculated considering that the major findings of this paper are the new estimates of the riverine carbon loads. Detailed explanation is needed on how river flow was measured. The method of load estimation appears to be too simple and with many assumptions, not specifying errors associated with each step. There are several methods for load estimation (e.g. Sickman, J.O. et al., 2007, Water Resources Research, Effects of urbanization on ...) you may try these and compare the results because load calculation is crucial to draw conclusions. One way to calculate daily load of stream ions is to use the LOADEST software developed by USGS if daily water discharge data are available. The software also provides confidence intervals.

Reply: Many thanks for your comment. Estimating riverine carbon flux is a very important part of this study in which we attempt to investigate the fate of carbon after entering the drainage network from terrestrial ecosystems. Just as you have pointed out, there are a number of methods to estimate the annual fluxes of dissolved and particulate matter transported by rivers. Major methods currently used include linear interpolation and ratio estimators, regression-based methods historically employed by the USGS, and recent flexible techniques such as Weighted Regressions on Time, Discharge, and Season (WRTDS), etc. As you have also suggested, the most commonly used USGS software package for estimating constituent load using regression is known as LOADEST (Runkel et al., 2004. Load Estimator (LOADEST): A FORTRAN Program for Estimating Constituent Loads in Streams and Rivers. U. S. Geological Survey Techniques and Methods Book 4, Chapter A5). Lee et al. (2016) recently reviewed the potential for flux estimation bias across a broader range of estimation methods and concluded that the Beale's ratio estimator and WRTDS generally exhibit greater estimation accuracy and lower bias (Lee et al., 2016. An evaluation of methods for estimating decadal stream loads. *Journal of Hydrology*, 542, 185-203). Our annual carbon flux estimation in this study was based on the Beale's stratified ratio estimator. Since the riverine carbon concentrations were measured with "sparse" sampling frequency while flow and suspended sediment had a continuous daily measurement, this method could greatly reduce the bias introduced by relatively low sampling frequency, in particular the high flow events that are often under-sampled (Parks and Baker. 1997. Sources and transport of organic carbon in an Arizona river-reservoir system. *Water Research*, 31, 1751-1759). Indeed, we have already used the Beale's ratio estimator in our earlier estimation of carbon flux in the Yellow River with success (i.e., Ran et al., 2013. Spatial and seasonal variability of organic carbon transport in the Yellow River, China. *Journal of Hydrology*, 498, 76-88). And the Beale's ratio estimator has proven to be highly reliable and is recommended if the relationship between discharge and concentration is weak (e.g., Fulweiler and Nixon, 2005. *Biogeochemistry*, 74, 115-130; Awad et al., 2017. *Environmental Pollution*, 220, 788–796; Chen et al., 2014. *Journal of Geophysical Research: Biogeosciences*, 119, 95-109; Sun et al., 2017. *Hydrological Processes*, 31, 2062-2075). In comparison, we have also estimated the carbon flux by using the suggested LOADEST software package. The flux results show high consistency with each other, with a difference of less than 4.5%. We have added a detailed description of the estimate method (i.e., the Beale's ratio estimator) in the revised manuscript. Please refer to the highlighted changes in the text. (lines 161-180)

Lines 160-: Do you mean the POC concentration not "content"? It appears the term "content" is misused throughout the manuscript.

Reply: Based on your and other reviewers' comments, we have re-defined the concentration of POC throughout the manuscript. It should be the POC concentration (POC%) in the total suspended solids (dry weight). (lines 161-163)

Lines 170-: How large is the river width? If it is near or lower than 90 meters, how can you estimate river surface areas using the DEM data of 90-m resolution? In other words, aren't you using too coarse data to estimate river water surface areas?
More detailed explanation is required on how the water surface area is calculated since this is a critical term for CO2 evasion estimates.
Reply: Because the Wuding River catchment is located in an arid-semiarid climate zone, the rivers and streams of drainage network is generally narrower than their counterparts in tropical rivers due to lower water discharge. The widths of the rivers and streams vary from 1.8 (first order streams) to ~61 m (the mainstem channel) (see Table 1 in our earlier wor: Ran et al., 2017. *JGR-Biogeosciences*, 122, 1439-1455), significantly lower than the DEM resolution of 90 m. Therefore, we only used the DEM data to delineate the drainage network in terms of stream length (usually >2.5 km) and stream number. The delineated drainage network was also calibrated through ground truthing during our fieldwork. Because the width of all rivers is less than the resolution and it fluctuates between dry and wet seasons, we measured widths of all sampled rivers during our fieldwork and aggregated them based on stream order to calculate the water surface area. We have revised the description of the water surface area calculation in the revised manuscript. (lines 198-203)

Lines 183-198: The method is better than nothing for sure. However, it appears the used references are relatively old (1995 and 2000). Do you have newer references on heterotrophic respiration than those? How the errors associated with the approach are calculated?
Reply: To estimate the Wuding River catchment's net ecosystem production (NEP), we used the global soil $CO_2$ efflux database described by Raich and Potter (1995) and the heterotrophic soil respiration ($Rh$) estimated by Hanson et al. (2000). Based on the global soil respiration flux database (Raich and Potter, 1995), the $S_R$ for this catchment is the range of 400-500 g C $m^{-2}$ year$^{-1}$. Hence, we used 450±50 g C $m^{-2}$ year$^{-1}$ to represent its soil respiration. This rate is consistent with recent measurements under different vegetation types in this arid-semiarid region (e.g., Fu et al., 2013). Fu et al. (2013. Soil respiration as affected by vegetation types in a semiarid region of China. *Soil Science and Plant Nutrition*, 59, 715-726) measured total soil respiration in this arid-semiarid region. Their mean soil respiration rates under 4 different vegetation types are in the range of 1-1.4 µmol $m^{-2}$ s$^{-1}$, which are equivalent to 380-530 g C $m^{-2}$ year$^{-1}$. Thus, our estimate is reliable. Although the references are relatively old, using the ratios derived from Hanson et al. (2000) has been widely used to assess heterotrophic soil respiration in river catchments under different land cover types (e.g., Brunet et al., 2009. Terrestrial and fluvial carbon fluxes in a tropical watershed: Nyong basin, Cameroon. *Chemical Geology*, 3, 563-572; Lee et al., 2017. A high-resolution carbon balance in a small temperate catchment: Insights from the Schwabach River, Germany. *Applied Geochemistry*, 85, 86-96). In addition, the propagated errors are calculated and presented in the revised manuscript. We have also added new references to justify our arguments. (lines 316-326)

Lines 208-: Does the 'sediment' mean 'suspended sediment'? If so, please clarify it to prevent confusion. Do you mean the POC concentration not "content"?

Reply: Yes, here it means the POC concentration in suspended sediment. We have clarified this in the revised manuscript 'The POC% in suspended solids…'. (lines 161-163)

Lines 222- and throughout the manuscript: What is the "+/-"? Standard deviation? Or standard error?
Reply: The '±' denotes standard deviation (SD) throughout the manuscript. We have explicitly indicated this when it is used for the first time in the revised manuscript (i.e., in Figure 2). Many thanks.

Lines 225-: While [DOC] (3.3 mg/L) is larger than [POC] (0.61 mg/L), the DOC export is much lower (0.3*10^10 g (yr-1?)) than POC export (3.7*10^10 g (yr-1?)). Why is that so?
Reply: Here the DOC concentration (mg/L) is expressed as the DOC content per unit volume of water, and the POC is expressed as POC% in total suspended solids (TSS, dry weight). Although the DOC concentration is larger than the POC%, the annual water discharge ($7.71 \times 10^8$ m$^3$/yr) at the catchment outlet Baijiachuan gauge is relatively low due to low precipitation and the concomitant annual TSS flux ($610 \times 10^{10}$ g/yr) is quite high owing to severe soil erosion. As a result, the annual DOC flux (g C/yr) is much lower than the POC flux.

Lines 228-233: The river water discharge and carbon loads can be highly dependent on precipitation. Was the year of field campaign categorized as wet, dry, or normal year compared to the long term mean (e.g. 1980-2017 precipitation)?
Reply: The multiannual precipitation for the Wuding River catchment is in the range of 300-500 mm during the period 1956-2010 with a mean precipitation of 430 mm/yr (available at http://www.yellowriver.gov.cn/; Li et al., 2007. *Hydrological Processes*, 21, 3485-3491). The precipitation in 2015 is about 410 mm, indicative of a normal year relative to the long-term mean precipitation. In comparison, the precipitation in 2017 is larger than 540 mm, significantly higher than the long-term mean precipitation (i.e., 26% higher). That is why we used the 2015 hydrological data to calculate the carbon flux. Another reason is because the three seasonal samplings were also performed in 2015. We have revised the hydrological information in the manuscript. (lines 256-261)

Lines 258–261: As the authors mentioned, the precipitation is high during summer. Thus, this assumption of no significant seasonal fluctuations may not be valid. Can you provide a range of stream surface area and CO2 evasion depending on season?
Reply: For $CO_2$ evasion from river waters, we separately estimated the total water surface of rivers in spring, summer, and autumn (please refer to Table S4 in Supplement for the estimated water surface area in the three seasons) and calculated the $CO_2$ evasion in these three seasons. The annual total $CO_2$ evasion was obtained by summing up the three seasonal $CO_2$ estimates. But for $CO_2$ evasion from reservoir waters, because these check dam-formed reservoirs are mostly constructed in steep gully channels and operated primarily for the purpose of sediment trapping and water storage, variation of the water surface area is much less significant than that of the rivers. Although there are also seasonal fluctuations, the magnitude should be quite minor compared with rivers. Thus, we assumed that there was no significant seasonal variation. (lines 289-293)

Lines 300–: Is the decreasing trend of DOC (Fig. 2) statistically significant? It appears the error bars are large. If this is not statistically significant, the following argument is vague. The decrease of DOC can be microbial- or photo-degradation to CO2, sorption to particulate matter, and dilution from increased water discharge of low [DOC]. The following discussion is speculative and could be strengthened by checking each factor.

Reply: Based on your comment, we have performed the significance test for DOC concentrations along the stream order. Because of the large error bars as shown in the figure, the decreasing trend of DOC is not statistically significant at the 95% confidence level. To reflect the downstream DOC concentration change, we aggregated the 1st-5th streams into 2 groups, including the headwater 1st-2nd streams and the higher order 3rd-5th streams, because it is usually believed that headwater low-order streams process organic carbon more rapidly and emit $CO_2$ at faster rates than downstream high-order streams (e.g., Butman and Raymond, *Nature Geoscience*, 4, 839-842; Crawford et al. 2013. *Journal of Geophysical Research: Biogeosciences*, 118, 482-494). Our results indicated that the DOC concentrations in the headwater 1st-2nd streams were on average 16-39% higher than that in the downstream 3rd-5th streams. Thus, the downstream DOC decline in the 1st-5th order streams likely suggests the mineralization of the bioavailable fraction of DOC along the river course (Figure 2), especially in spring and autumn. In addition, sorption and input of increased water with low DOC may partially dilute the DOC concentration as you commented. However, in view of the spatial homogeneity in terms of soil erosion rate, SOC content in soils, and hydrologic regime within each subcatchment and the spatially constant POC% from the headwater to the mainstem channel, the sorption and 'dilution effect' are expected to be minimal. Accordingly, we have revised the claims in the manuscript. (lines 230-245; 333-349)

Line 325–336 (and lines 385–394, and Fig. 6): I am confused. Do you mean the POC concentration not "content"? Why the "content" has the unit of concentration, %, not just grams? I think heavy rain during summer could generate high POC content but low POC concentration. Please clarify.

Reply: Many thanks for your comment. It should have been POC concentration in the text. We have clarified the POC concentration in the total suspended solids (TSS, dry weight) in the units of POC% throughout the manuscript. By multiplying the annual TSS flux, we can calculate the annual POC flux.

Line 351–368: The pCO2 is a function of pH and alkalinity. The pCO2 is high when the water pH is low. The ground water of the area has the pH of >~8. Then, the calculated pCO2 is very low which is well described in the line 211. Then, how CO2 evasion can be high when pCO2 is low? Please clarify.

Reply: Just as you have pointed out, $pCO_2$ is a function of pH and alkalinity, and it can be calculated from the latter two variables. The observed pH in the study catchment ranged from 7.68 to 9.29 and the pH in groundwater is generally slightly higher than 8.0. Even so, for the sandy subcatchment reservoirs, the pH of the groundwater is still lower than that of the river water into the reservoirs (e.g., 8.7-9.3). With extremely high alkalinity (DIC) concentrations, despite the relatively high pH of around 8.0, the calculated $pCO_2$ is well above the atmospheric equilibrium (i.e., ~390 µatm), and facilitates the observed $CO_2$ evasion. We have revised the manuscript to make the claim more clear and accurate. (lines 398-405)

Lines 400: Which part of the Figure S1 supports this sentence?
Reply: We have added the information on carbon export during typical floods in the Supplement (Figure S1).

Lines 430-481: Very interesting findings.
Reply: Many thanks for your comments. We collected carbon isotope samples of the emitted $CO_2$ from river waters and attempted to explore its potential sources in association with carbonate dissolution and respiration of recent organic matter.

Tables: What is the "+/-"? Standard deviation? Or standard error?
Reply: The '±' denotes standard deviation (SD) and the description has been added in the revised manuscript.

Table 1: Please provide information on how many reservoirs were used to draw the table.
Reply: There are currently 337 reservoirs in operation within the Wuding River catchment (please see the figure below). This information has been added into the caption, and this map has also been included in the Supplement (Figure S2).

[Figure]

Figure: Spatial location of the 337 reservoirs within the Wuding River catchment.

Figure captions need to provide more detailed description of the figures including explanation on legends.
Reply: We have significantly improved the figure captions based on your comments and detailed information has been added. Please refer to highlighted changes in the revised manuscript.

Figure 1: It is hard to differentiate the colors of the stream order, especially with the background altitude colors. Please revise the figure so that each symbol can be seen clearly.
Reply: We have carefully adjusted this figure in terms of color scheme, marker size, label size, etc., and have added the subcatchment boundaries to make the figure much easier to read.

---

## Author Comment (AC2) · 16 Apr 2018

Dear Dr Hemingway,

We thank you very much for your comments on our manuscript. Based on your very constructive comments, we have thoroughly revised the manuscript. Additional discussion and justifications have been added into the manuscript or into the Supplement. Please see below the detailed responses. Major changes have also been highlighted in the revised manuscript.

With best regards
Lishan Ran, on behalf of the coauthors
* * *
Synopsis
The central focus of this manuscript is to investigate carbon cycling in the arid-semiarid Wuding River catchment using both campaign-style and time-series sampling approaches. The authors quantify dissolved carbon concentrations, both organic (DOC) and inorganic (DIC), as well as particulate organic carbon (POC) concentrations and CO2 outgassing fluxes throughout the catchment over multiple seasons. In particular, the authors compare and contrast signals across a range of Strahler stream orders (1 to 6) from subcatchments underlain by sand and by loess and quantify differences in their respective carbon budgets.

As the authors point out, arid-semiarid river catchments are severely underrepresented in global riverine carbon-cycle budgets. By presenting a large dataset for the Wuding River catchment, this study begins to ameliorate this issue. I therefore find the goals and targets of the present study to be impactful, as they attempt to advance our collective understanding riverine carbon cycling. However, I do have some issues with the interpretation of these data, particularly related to a number of claims that seem unsubstantiated or somewhat contradictory. Additionally, I feel that there are some areas that warrant further clarification and detail. Overall, I feel that the authors should remove some of the weaker and highly speculative text that attempts to prescribe carbon sources and should instead focus on the strengths of this dataset – namely, carbon fluxes and budgets. If the authors can address these issues, which I think they can, then I believe that this manuscript could provide a valuable contribution to Biogeosciences.

I outline my larger concerns in detail below, followed by a list of smaller concerns and questions. Please do not hesitate to contact me for further discussion regarding this review.

Sincerely,
Jordon Hemingway
jordon_hemingway@fas.harvard.edu
Reply: Many thanks for your very constructive comments. Please find below our responses to each of your comments.

Larger Comments
Methods details and measurement uncertainty
In general, I feel that more detail is required in describing the methodology and presenting data uncertainty. In particular, the paragraph beginning on L135 should be expanded considerably. For example, I would like to see more details related to:
Reply: Many thanks for your very constructive comments.

i) Field titration methods. How was this done? Were any standards measured? Field titrations generally have quite high uncertainty associated with them (~5 – 10%), yet there is no uncertainty assessment presented here. What is the resulting propagated uncertainty for calculated DIC concentration values?

Reply: Total alkalinity was determined by triplicate titrations in the field with 0.1 M HCl, and methyl orange was used as the indicator, following the standards as suggested by APHA (1999, *Standard Methods for the Examination of Water and Wastewater*). For the Wuding River with widespread presence of carbonates, its river water alkalinity is quite high (62.1–67.7 mg $L^{-1}$). Our field triplicate titration results are highly consistent with the difference between the three results generally less than 3%. Thus, we expected the obtained alkalinity results are reliable with high confidence. Finally, DIC was calculated from total alkalinity, pH, and temperature by using the program CO2calc. Because the measured pH varied from 7.68 to 9.29, the calculated DIC was approximately equal to alkalinity, with >96% of the alkalinity composed of $HCO_3^-$, consistent with the relative speciation (%) of $CO_2$, $HCO_3^-$, and $CO_3^{--}$ in water as a function of pH (please refer to the figure below). The revised descriptions have been added into the manuscript. (lines 142-145)

[Figure]

Figure: Relative concentrations of the different inorganic carbon compounds against pH.

ii) DOC uncertainty. How was DOC uncertainty estimated? Was a standard calibration curve used? If so, how often was the calibration curve analyzed? Was each sample injected in triplicate? Duplicate? Single injection?

Reply: DOC was determined by the high-temperature combustion method (850 ℃) by using an Elementar Vario TOC Select Analyzer. A standard calibration curve was used for every round of field samples. Generally, the standard calibration curve was analyzed and re-determined for each 60-80 samples, depending on the variability of the DOC concentration. Triple injections indicated an analytical precision of <3%, and the average of the three injection results was calculated to represent the sample's DOC concentration. These descriptions have been added into the revised manuscript. (lines 140-142)

iii) Were solid samples fumigated with HCl at room temperature or at ≥60°C? I ask because dolomite will not be removed at temperatures below 60°C. If these samples are expected to contain dolomite, and if they were fumigated at room temperature, then I would expect resulting POC estimates to be biased upward.

Reply: To measure the POC concentration, the solid soil and sediment samples were fumigated at 65 ℃ for 24 h. For the Chinese Loess Plateau, carbonates in its loess–paleosols consist mostly

of calcite and dolomite, and the latter is the primary detrital material (please see Yang et al., 2000. *Palaeogeography, Palaeoclimatology, Palaeoecology*, 157, 151-159). Therefore, we carefully removed the dolomite with concentrated HCl at a higher temperature than the room temperature. We have added the description into the revised manuscript. (lines 149-151)

iv) CO2 d13C values. Were these analyzed by Beta Analytic using an IRMS on a separate gas split, or are these values generated by the AMS? I would expect these to be IRMS values, but this should be stated clearly.

Reply: The $CO_2$ $\delta^{13}C$ results are generated by the AMS at the Beta Analytic Radiocarbon Dating Laboratory (Miami, USA). This has been clearly stated in the revised manuscript. (lines 156-158)

v) Radiocarbon notation. Throughout the manuscript, the authors conflate 14C age, Δ14C (which is always reported in units of per mille!) and percent modern, or pMC. I would strongly suggest that the authors choose one notation and stick with it (my personal choice would be to use pMC). Still, if the authors choose to use 14C age, this be reported in units of "14C yr BP" rather than simply "years", as the latter is ambiguous and could refer to a calibrated age, which would not be appropriate here.

Reply: Thanks a lot for your suggestion on how to describe the $^{14}C$ analysis results. We have chosen to use the percent modern (pMC) to describe the results throughout the manuscript, mainly in Sections 3.2 and 4.3. But, to compare our results with Wang et al (2012. *Global Biogeochemical Cycles*, 10, 26, GB2025, doi:10.1029/2011GB004130) that investigated the $^{14}C$ age of DOC and POC in the Yellow River, we have kept '$^{14}C$ age' results in Table 2 and Figure 5, and the simple notations 'years' have been replaced by '$^{14}C$ yr BP'. (lines 153-156; 533-540)

vi) Sediment accumulation rates. In Figure 10, a burial flux is presented in units of g C yr-1, yet I find no reference to calculations for sediment accumulation rates (SAR). How was SAR calculated for each of these cores? This information is necessary in order to convert the measured %OC numbers into burial fluxes…

Reply: The sediment accumulation rate behind check dams in the Wuding River catchment was based on our earlier estimate (i.e., Ran et al., 2013. *Global and Planetary Change*, 10, 308-319; please also refer to Section 3.3 of the manuscript). Our earlier work shows that the annual sediment accumulation rate in this study catchment is $3720\times10^{10}$ g year$^{-1}$. In addition, based on the POC concentration (POC%) of the four sediment cores distributed in both the sandy and loess subcatchments (Figure 1), we calculated the arithmetic mean of the POC% (0.21±0.11%). With the sediment accumulation rate and the POC% in deposited sediment we estimated the total OC burial rate (($7.8\pm4.1)\times10^{10}$ g C year$^{-1}$). Because the POC content in the top 0-60 cm soils is considerably higher that that in the deeper soil layers, our simple estimate is associated with great uncertainty. Future efforts are therefore needed for a more accurate assessment. In addition to Section 3.3, we have also added these justifications into the manuscript. (lines 469-477)

Additionally, all of the numbers reported in the "Results" section should include corresponding uncertainty, either analytical uncertainty (when reporting single values) or sample population uncertainty (when presenting averages). For averages, please be clear if reporting standard errors or standard deviations. Similarly, significant figures should be consistent throughout the manuscript!

Reply: Many thanks for your suggestion. We have provided the uncertainty, mainly standard deviation, for all of the numbers in the 'Results' section. In addition, we have also double checked the consistency of significant figures throughout the manuscript. Please refer to highlighted changes in the revised manuscript. (lines 230-245; 261-265; 289-298; 316-326)

Net Ecosystem Production

I am left somewhat confused by the assumptions and uncertainties related to NEP calculations. To convert SR to Rh, the authors apply a "forested" and "non-forested" fraction heterotrophic derived from Hanson et al. (2000). However, the "non-forested" estimates from this reference are for pasture and grassland, not barren landscapes such as those presented in the current study. Presumably nearly 100% of soil respiration on barren landscapes is heterotrophic, no? Additionally, while the "non-forested" fraction heterotrophic in Hanson et al. averages 40%, they observe values ranging from 10% to 90% -- nearly the entire possible range!

I wonder if the authors have any way to estimate the uncertainty on NEP estimates presented here – if so, these should be discussed in detail. I would expect these uncertainties to be quite large, yet this is not mentioned or discussed in the manuscript. For example, how do the values here compare to those calculated by subtracting SR from MODIS-derived GPP values? To me, this seems like a more straightforward method to estimate NEP that isn't subject to the uncertainties associated with converting SR to Rh.

Reply: We divided the study catchment into two subcatchments, including the sandy subcathment and the loess subcatchment. While forest cover in the Wuding River catchment is quite low (less than 5%) as a result of low precipitation, grassland is the major land cover in the sandy subcatchment and agriculture and grassland predominate the loess subcatchment (Wang et al., 2014. Spatial-temporal changes of land use in Wuding River Basin under ecological restoration, *Bulletin of Soil and Water Conservation*, 34, 237-243 (in Chinese with English abstract). This is largely the result of the implementation of the Grain-for-Green Project which was initiated by the Chinese government in 1999. After more than 10 years of implementation of this vegetation restoration program, the vegetation cover (forest and grassland) has greatly increased. Please also refer to two photos below showing the landscape of the sandy subcatchment (left) and of the loess subcatchment (right). Both photos were taken by me in 2015 when doing the fieldwork. To better describe the landscape of the catchment, we have revised the description in Section 2.1 'Study area' (lines 88-90). Therefore, the landscape and land cover of the Wuding River catchment are generally consistent with the distinction of "forested" and "non-forested" by Hanson et al. (2000). With respect to the huge range of the "non-forested" fraction heterotrophic (i.e.,10-90% as you have noticed), we have discussed the potential uncertainty in the revised manuscript. (lines 560-563).

[Figure]

Figure: Landscape characteristics of the sandy (left) and loess (right) subcatchments.

Our rate is consistent with recent measurements under different vegetation types in this arid-semiarid region (e.g., Fu et al., 2013). Fu et al. (2013. Soil respiration as affected by vegetation types in a semiarid region of China. *Soil Science and Plant Nutrition*, 59, 715-726) measured total soil respiration in this arid-semiarid region. Their mean soil respiration rates under 4 different vegetation types are in the range of 1-1.4 µmol m$^{-2}$ s$^{-1}$, which are equivalent to 380-530 g C m$^{-2}$ year$^{-1}$. Thus, our estimate is reliable. We have carefully revised the manuscript with new references to justify our arguments (lines 316-326). Using the ratios derived from Hanson et al. (2000) has been widely used in the world to assess heterotrophic soil respiration in river catchments under different land cover types (e.g., Brunet et al., 2009. Terrestrial and fluvial carbon fluxes in a tropical watershed: Nyong basin, Cameroon. *Chemical Geology*, 3, 563-572; Lee et al., 2017. A high-resolution carbon balance in a small temperate catchment: Insights from the Schwabach River, Germany. *Applied Geochemistry*, 85, 86-96). Just as you have commented, this portioning is associated with potential uncertainty. Our ongoing research assessing NEP storage dynamics on the entire Loess Plateau is using the MODIS-derived GPP products. A preliminary estimate for the Wuding River catchment suggests that the results of the two methods are generally equal with a difference of ~11%. We greatly appreciate your suggestion and we will adopt the more straightforward method. Many thanks.

Interpretation of DIC, CO2 d13C, and Δ14C
    In general, I am confused by the discussion on DIC sources, especially as they relate to measured CO2 d13C and Δ14C values – there seem to be a number claims that are either contradictory or are not explained in significant detail. Beginning in the abstract (L21) and repeated throughout the manuscript, the authors state that DIC is largely sourced from carbonate dissolution, especially in the loess subcatchment. Intuitively, this makes sense to me since loess contains a significant amount of carbonate, as the authors rightly state. However, this is incompatible with the d13C and Δ14C values presented in this study, which suggest that remineralization of terrestrially derived OC is the main source of outgassed CO2 behind check dams. What mechanisms could explain this discrepancy? I feel that there needs to be significantly more discussion and clarification here.
    Furthermore, I find some of the claims related to CO2 outgassing to be overstated. For example, the statement: "The evasion of old carbon [derived from pre-aged OC respiration as is seen here] is likely to be widespread in arid-semiarid catchments worldwide with similar hydrological regime and terrestrial ecosystems" (L477). This seems to be quite a stretch, especially given my confusion related to the lack of carbonate dissolution signature as stated above.
Reply: We collected $CO_2$ emission samples in the Wuding River catchment for carbon isotope analysis by using the $SrCl_2$ solution. Thus, the measured $\delta^{13}C$ and $\Delta^{14}C$ results are for the emitted $CO_2$ from river water. Unfortunately, we did not collect water samples for $\delta^{13}C$ and $\Delta^{14}C$ analysis of the DIC. But prior studies indicate that the $\delta^{13}C$ of DIC generally ranges from -6.7‰ to -12.9‰ in Loess Plateau rivers (Liu and Xing, 2012. *Chemical Geology*, 296, 66-72). For the arid-semiarid Wuding River catchment and the whole Yellow River basin in which the Wuding River is located, carbonate dissolution has been found to be the primary source of DIC (mainly $HCO_3^-$) due to its high carbonate content in loess soils (up to 20%; please see Chen et al., 1995. Major element chemistry of the Huanghe (Yellow River), China: Weathering processes and

chemical fluxes. *Journal of Hydrology*, 168, 173-203; Chen et al., 2005. Spatial and temporal analysis of water chemistry records (1958–2000) in the Huanghe (Yellow River) basin. *Global Biogeochemical Cycles*, GB3016, doi: 10.1029/2004gb002325). Therefore, we conclude that DIC is largely sourced from carbonate dissolution, especially in the loess subcatchment. With respect to $CO_2$ emissions, however, the emitted $CO_2$ is characterized by much depleted $\delta^{13}C$ values (-19.3‰ – -33.9‰), which is significantly different from the $\delta^{13}C$ signature of DIC carbonates (i.e., 0‰ for DIC derived from carbonates by proton attack and -8.5‰ for DIC derived from carbonate dissolution; Barth et al., 2003. *Chemical Geology*, 200, 203-216; Brunet et al., 2009. *Chemical Geology*, 265, 563-572). In comparison, the $\delta^{13}C$ values of the emitted $CO_2$ largely reflect the contribution of C3 and C4 plants which have a $\delta^{13}C$ values signature of -27‰ and -15‰, respectively. Mineralization of terrestrially derived OC has been widely found to be the primary source of river water $CO_2$ emissions. For example, Mayoga et al. (2005) found that respiration of contemporary organic matter (less than 5 years old) originating on land and near rivers is the dominant source of excess $CO_2$ that drives outgassing in the Amazon rivers (Mayorga et al., 2005. *Nature*, 436, 538-541). Similarly, Borges et al. (2015) discovered that lateral transport of soil or wetland DOC and POC that is mineralized to $CO_2$ within the rivers maintains $CO_2$ outgassing in African rivers (Borges et al., 2015. *Nature Geoscience*, 8, 637-642). Therefore, in combination with the measured $\delta^{13}C$ and $\Delta^{14}C$ results of the emitted $CO_2$, we conclude that decomposition of the terrestrially derived OC drives $CO_2$ outgassing in the Wuding River catchment although DIC is largely originated from carbonate dissolution. We have added more justifications with relevant references in the revised version of the manuscript to support our arguments. Based on your comments, we have removed the overstated comments, including the one you mentioned here, from the revised manuscript to make sure all the arguments are supported by our results and figures. Major changes have been highlighted in the manuscript. (lines 495-504; 525-528; 533-540)

DOC sources and trends

Beginning on L204 and continuing throughout the manuscript, the authors refer to a "significant downward trend along the river course from headwater downstream… in both subcatchments." However, when I look at Figure 2, I am left puzzled and wondering if these trends are, in fact, significant. Given the large error bars for each stream order, my guess is that they are not. In my opinion, any subsequent discussion related to DOC sources and trends (e.g. L300-313; L306-309; L318-324) is highly speculative at best.

Additionally, I find some of these claims to be contradictory. For example, on L314, the authors state that "…there was no significant correlation between DOC and flow based on the spatial sampling results". However, for the high-frequency sampling the authors observe a "positive correlation between DOC export and hydrography [that] demonstrates the enhanced leaching of organic matter from surface vegetation and organic-rich top soil layers". Why would a positive correlation be expected during storm events yet not on a seasonal basis? What mechanism could explain this? This discrepancy is not addressed.

Reply: Many thanks for your comment on DOC sources and trends. To detect the DOC concentration changes along the river course from headwater downstream, we plotted the average DOC concentration with standard deviation (error bars) based on stream order (Figure 2). The 'DOC first exhibited a downward trend along the river course from headwater downstream and then increased in the 6th order mainstem river in both the sandy and loess subcatchments (Figure 2)'. Because the downward trend does not pass the significance test at the

significance level of 0.05, we did not use the word '*significant*' in the description. If we categorize the first 5 stream orders (1–5) into 2 groups (1st–2nd and 3rd–5th), we can easily detect that DOC in the headwater 1st–2nd order streams (4.7–5.4 mg L$^{-1}$) was on average 9  21% higher than in the 3th–5th order streams (4.2–4.9 mg L$^{-1}$), it increased to 5.2–6.1 mg L$^{-1}$ in the 6th order mainstem, representing an increase of 18–36% relative to the 3th–5th order streams. This is particularly true for the loess subcatchment (Figure 2a). When combining the two subcatchments together, the DOC in the 6th mainstem was 5.7 mg L$^{-1}$, which was 27% higher than the average of the 3rd–5th order streams (4.5 g L$^{-1}$). To more accurately describe the DOC trend, we have revised the statement: 'Although statistically insignificant, DOC first exhibited a downward trend along the river course…', and also the description of the results (lines 230-237).

When plotting the DOC measured across the whole catchment over three seasons against the concomitant flow, there was no significant correlation between DOC and flow based on the spatial sampling results (p>0.05; please refer to the graph below). In comparison, our high-frequency sampling at the catchment outlet Baijiachuan gauge indicates that DOC concentrations were 26% higher in the flooding periods than that in normal flow conditions. The positive correlation between DOC export and hydrography demonstrates the enhanced leaching of organic matter from surface vegetation and organic-rich top soil layers (Hernes et al., 2008. *Geochimica et Cosmochimica Acta*, 72, 5266-5277). Clearly, this positive response contradicts the indiscernible relationship between DOC and flow discharge within the catchment. This is probably because the three intensive seasonal samplings did not capture the carbon export in high-flow conditions. The flow discharge during the three sampling periods varied in the range of 0.002–105 m$^3$ s$^{-1}$ (please see this range in the figure below), which largely reflects the carbon export processes during low flow to, at most, medium flow conditions. In comparison, the high-frequency sampling at Baijiachuan gauge captured the carbon export during extremely high flows (200–1760 m$^3$ s$^{-1}$). We have also presented the raw data of monthly flow discharge and DOC concentrations in the Supplement (Table S2), so they are now available for free use. Please refer to the explanation of this discrepancy in the revised manuscript (lines 417-429).

[Figure]

Figure: Relationship between flow discharge and DOC based on the three sampling results.

POC sources and trends

I find that a significant amount of discussion related to POC sources and sinks needs to be substantiated with more evidence or, at a minimum, alternate explanations need to be addressed. First, beginning on L326, the authors claim that low POC content (by which they mean % of suspended solids, a point that I address below) "reflects the ancient sedimentary OC origin of about 0.5% for fluvial sediments worldwide… [and is also] seen from the isotopic signature of the Yellow River sediment…" The authors go on to state that low %OC reflects "mobilization of subsurface soils that have a substantially lower OC content than surface soils" (L334). However, "ancient sedimentary OC" presumably refers to sedimentary rock derived material, which is certainly not the same as "subsurface soils". I'm left confused as to what the authors expect to be the major source of POC – sedimentary rocks or subsurface soils? I think that, with concentration measurements alone, one cannot make strong claims either way.

The well-known relationship between grain size and %OC is also not addressed. The observed POC concentration trends could easily be explained by variable hydrologic sorting – i.e. coarser, OC-poor sediments that are transported during high discharge periods – which would mask any POC source signal. In the absence of isotopic (d13C, Δ14C) or grain-size-dependent measurements (e.g. %OC as a function of Al/Si ratios), I find it hard to believe that POC sources can be prescribed as is done here (also repeated beginning on L385).

Reply: Thanks a lot for your comment. Just as you mentioned, in this manuscript we expressed the POC content in the total suspended solids (TSS). Therefore, it is a percentage of the TSS (dry weight). To make it more clear and consistent throughout the manuscript, we have replaced this term with 'POC%' throughout the manuscript, including all figures and tables. This has been explicitly introduced in manuscript (lines 161-163). As for the sources of POC, it is closely related to the soil erosion and sediment yielding characteristics of the Wuding River catchment, or generally, the whole Chinese Loess Plateau. The Chinese Loess Plateau (area: ~440,000 km$^2$) is covered with 100-300 m thick highly weathered loess soils (Zhao et al., 2013. *Land Degradation & Development*, 24, 499–510; Nie et al., 2015. *Nature Communications*, 6:8511, doi: 10.1038/ncomms9511). As a result of the very fine soil particles, soils in the Loess Plateau are extremely susceptible to erosion. And gully erosion is the major erosion type and is responsible for >70% of the total erosion rate for most parts of the Loess Plateau (Xu, 1999. *Catena*, 36, 1-19; Li et al., 2015. *Geomorphology*, 248, 264-272). Actually, gully erosion of tens of meters is quite common (please also see the photo below for a visual experience).

[Figure]

Figure: Gully erosion on the Loess Plateau.

The low POC% in the sampled sediments is quite close to the organic carbon content of sedimentary rocks (i.e., 0.5%; Ludwig et al., 1996. *Global Biogeochemical Cycles*, 10, 23-41). Recent studies investigating POC of the Yellow River sediment by means of $\delta^{13}C$ and $\Delta^{14}C$ analysis also suggest that its POC is not from the recently fixed terrestrial plant materials and freshwater plankton, but from the highly decomposed loess soils and weathering of sedimentary rocks and ancient kerogen (Wang et al., 2012. *Global Biogeochemical Cycles*, 10, 26, GB2025, doi:10.1029/2011GB004130). As the primary source of the Yellow River sediment, we can expect that the sediment in the Wuding River catchment carries similar carbon isotopic signatures as that in the Yellow River sediment. Therefore, in association with the high contribution of gully erosion to annual TSS transport and the low organic carbon content of the sampled TSS, we concluded that the lower POC% in suspended solids in summer likely reflects the origin of sedimentary rocks mobilized by gully erosion. Based on your comment, we have carefully revised the manuscript and corrected the misuse of 'subsurface soils' and 'ancient sedimentary rocks'. In addition, necessary references have been added to justify the arguments. Please refer the highlighted changes in the manuscript (lines 369-378).

With respect to the relationship between grain size and POC%, prior studies have investigated the POC% changes in relation to the grain size of sediment in the Loess Plateau and the Yellow River. For example, Zhang et al. (2013. *Biogeosciences*, 10, 2513–2524) divided TSS into five categories (i.e., <8 µm, 8–16 µm, 16–32 µm, 32–63 µm, and >63 µm) and determined the POC% of each category. Their results show that more than 75 % of the POC was concentrated in sediment particles with grain size smaller than 16 µm, which suggests that the TSS grain size was the dominant factor controlling POC transport in the Loess Plateau and the Yellow River. Same results of a higher POC% in smaller particles are also discovered by Wang et al. (2012. *Global Biogeochemical Cycles*, 10, 26, GB2025, doi:10.1029/2011GB004130). We have added these justifications into the revised manuscript, and references have been used to justify our arguments (lines 434-446). In addition, our ongoing (biweekly) sampling at the catchment outlet Baijiachuan gauge is aimed to explore the relationship between grain size and POC%, and hopefully the sources of POC could be better prescribed. Many thanks for your comments and inspirational suggestions.

Similarly, beginning on L408, %OC content behind check dams is compared to that on hillslopes and is used as evidence for burial efficiency. However, this "negligible OC loss after erosion" (L412) could be explained by alternative hypotheses. For example, deposited material could (likely does?) contain a different grain size distribution than that of hillslope soils, and thus a different %OC content. Also, any remineralization of terrestrially derived POC could be masked due to replacement by aquatic sources (as is discussed). Again, I find it hard to prescribe POC sources and burial efficiencies without additional measurements such as d13C and Δ14C. I also find the claim that this material "would have otherwise been mineralized to form CO2 or CH4 along fluvial delivery" (L418) to be somewhat speculative. Presumably some of this material would have been transported and buried in coastal marine sediments. Heuristically, it makes sense that burial efficiencies behind check dams are higher than for coastal marine sediments, as the authors imply, but I find a general lack of evidence supporting this claim. Reply: The hilly areas of the Loess Plateau is dominated by gully erosion which can mobilize both the surface soils and the deeper soils as shown in the figure above. And numerous studies on soil erosion in the Loess Plateau have also confirmed the dominant role of gully erosion in

annual total soil erosion rate as mentioned above. For example, in the Wuding River catchment, Zheng et al. (2008. *Geomorphology*, 93, 288-301) concluded that '*when rainstorm intensity is sufficiently strong (> 0.3 mm min$^{-1}$), all grain-size fractions of loess on a hillslope are eroded without sorting*'. Because most of the check dams are constructed on gully channels and very close to the eroding sites, the eroded soils from hillslopes and gullies can be quickly trapped by check dams after a short delivery distance (usually less than 5 km based on our field surveys). The sediment trapping efficiency is surprisingly high (e.g., >90%) as most dams only have a small intake for irrigation and don't have spillway gates (please see a typical check dam shown below, taken by Lishan Ran during the fieldwork). Thus, the loss during fluvial transport is likely small as suggested by the comparison between POC% in sediments and soil OC in hillslopes. Moreover, just as you commented, any remineralization of terrestrially derived POC may have been masked due to replacement by aquatic sources. However, based on the combined use of $^{137}$Cs and $\delta^{13}$C techniques as well as C/N ratios, our earlier work (Wang et al., 2017. *Agriculture, Ecosystems and Environment*, 247, 290-297. Nufang Fang, a co-author of this manuscript, conceived the cited paper) discovered that most of the buried POC is derived from soil erosion from the catchment. In addition, approximately 70% of the eroded soil OC can be buried by check dams in the study catchment. Finally, eroded soil OC is subject to a number of biogeochemical processes, such as burial by impoundments, mineralization in the water column, outgassing, and export to the ocean depending on a suite of physicochemical conditions (e.g., Battin et al., 2009. *Nature Geoscience*, 2, 598-600; Drake et al., 2017. *Limnology and Oceanography Letters*, doi: 10.1002/lol2.10055). For the statement 'would have otherwise been mineralized to form $CO_2$ or $CH_4$ along fluvial delivery', we have reworded this sentence to make it more clear and accurate and added references to justify the claim. Please refer to the highlighted changes in the revised manuscript (lines 469-482).

[Figure]

Figure: A typical check dam in the Wuding River catchment.

Finally, I find that reporting "OC content" as %OC rather than a concentration (e.g. mg OC L-1) or a flux (e.g. t OC km-2 d-1) is ineffective and is somewhat misleading. For example, the authors state that "the substantially lower POC content in the wet season largely reflects the impact of gully erosion" (L385). However, one would expect that POC concentration and flux are actually significantly higher during the wet season! As described above, changes in %OC could reflect hydrologic sorting and are not necessarily indicative of source. I would strongly

recommend discussing POC trends in the context of concentration and flux, rather than %OC. This would allow the authors to shift the focus away from attempting to prescribe POC sources (which I find to be a weakness overall) and toward OC flux and budget estimates, which I think is a strength of this manuscript.

Reply: First of all, many thanks for your comment and suggestion. Because the POC sampling is conducted at 74 nested sites across the whole catchment (Figure 1), we did not delineate the boundary of the sub-catchment that each sampling site controls and calculate the POC yield (in units of t OC $km^{-2}$ $d^{-1}$) by normalizing to the size of each sub-catchment. Also, it is not feasible to calculate the annual POC flux at these sampling sites based only on the 3 sampling campaigns in spring, summer, and autumn. Instead, we only calculated the annual flux of C (g C $yr^{-1}$), including DOC, DIC, and POC, at the catchment outlet Baijiachuan gauge for which we have monthly C results and daily flow and sediment export data. And this gauge-based fluxes were used in the C budget to evaluate riverine carbon export in relation to NEP. To better present the POC results, we have now used POC% (i.e., the percentage of POC in total suspended solids (dry weight)) to express the POC content in suspended solids. The term 'POC%' has now been used throughout the manuscript to avoid unnecessary misunderstandings. Just as you have expected, although the POC% in the wet season is lower than that in the dry season, the POC flux in the wet season is considerable on an annual basis because of the high sediment loading, accounting for 65% of the annual total POC flux. Also, with respect to the potential sources of POC, we have thoroughly revised the manuscript based your earlier comments. Please refer to the highlighted changes in the revised manuscript (lines 434-447; 469-477).

Data availability

In my opinion, a major strength of data-rich manuscripts such as this is the ability for readers to incorporate these data into future studies – whether those be review articles or comparisons to other, similar catchments. Along those lines, I am left wondering why the authors do not make all of their raw data available as supplemental tables? I would strongly suggest do so or, at a minimum, including a "Data Availability" statement pointing the reader to a repository that includes these data.

Reply: Many thanks for your comment. We strongly agree with your suggestion. This study is an extension of our earlier work (Ran et al., 2017. *JGR-Biogeosciences*, 122, 1439-1455). In the Supplementary Information of the Ran et al. (2017) paper, we have already made most of our raw data used in this study available. These data include the physiochemical parameters (e.g., location, elevation, channel slope, flow velocity, wind speed, pH, water temperature, dissolved oxygen, Chl *a*, etc.), $CO_2$ emissions ($pCO_2$ and areal flux), and dissolved carbon concentration (DOC and DIC) in both river and reservoir waters. To facilitate future review studies and/or comparison analyses, we have made the leftover data available by presenting them in the Supplement of this study. Specifically, these data include POC of sediment samples (2015 and 2017) and of drilled sediment from check dams, monthly DOC and DIC concentrations at the catchment outlet (Baijiachuan gauge) as well as the concomitant flow information. Please refer to the Supplement for these data (Tables S1-S3).

Smaller Comments

L14: Remove dash between "terrestrially derived", change "represent" to "represents".

Reply: Changed.

L15 (also L68): What is meant by "redistribution"? Do the authors mean "partitioning between DIC, DOC, and POC"? I would change this wording for clarity.
Reply: Here we meant the fate of riverine carbon during its transport from headwater streams to the catchment outlet, including downstream export to catchment outlet, $CO_2$ evasion from water surface, and organic carbon (OC) burial through sediment storage. We have replaced the word 'redistribution' with 'fate' for clarity in both sentences. (Lines 15 and 69)

LL17: Change to "While DOC…"
Reply: Changed.

L18: What is meant by "DOC concentration is spatially comparable within the catchment"? I'm confused by this statement. Don't you argue that DOC concentrations decrease with increasing stream order? (although I question this trend, as stated above).
Reply: Based on your comment, we have removed this ambiguous claim and rephrased the abstract. We have also discussed the spatial variation of DOC concentration from the headwater streams to the mainstem channel. (lines 17-19; 231-236). Many thanks.

L19: "This reflects the enhanced…" seems overly confident. I would say "This likely reflects…"
Reply: Changed.

L21-22: I'm still confused by the DIC sources – carbonate dissolution seems incompatible with the measured CO2 d13C and $\Delta$14C values.
Reply: The measured $\delta^{13}C$ and $\Delta^{14}C$ values of the emitted $CO_2$ are different from that in the DIC of the Loess Plateau rivers (i.e., -6.7 to -12.9‰; Liu and Xing, 2012. Isotopic indicators of carbon and nitrogen cycles in river catchments during soil erosion in the arid Loess Plateau of China, *Chemical Geology*, 296, 66-72). Also, based on the $\delta^{13}C$ values of the DIC, Liu and Xing, (2012) discovered that it is largely derived from carbonate dissolution (48.1-94.6%,). The observed differences in this study reveal that the emitted $CO_2$ is not likely from carbonate dissolution-derived DIC. We have also revised the claim in the manuscript. (lines 495-504)

L23: Please be clear that you mean %OC in sediments when stating that "[POC content] shows low values in the wet season." As stated, this implies that POC concentration or flux are lower in the wet season, which I presume is not true.
Reply: Thanks a lot for your comment. We have clearly stated the 'POC%' in the abstract and throughout the manuscript.

L27 (and throughout): Please update the 14C notation, as described above. "Indicating the release of old carbon previously stored in soil horizons." Couldn't this also be described as a mixture of 14C-free carbonate dissolution and respired young OC? I'm not sure that this claim is supported.
Reply: Based on your major comment above, we have updated the $^{14}C$ notation throughout the manuscript. If looking at the $^{14}C$ results only, it could also a mixture of 14C-free carbonate dissolution and respired young OC as you suggested. But if we take the $\delta^{13}C$ results into account, it seems the contribution of carbonate dissolution is quite small, because the $\delta^{13}C$ signature of carbonate-derived DIC is 0‰ for DIC derived from carbonates by proton attack and -8.5‰ for DIC derived from carbonate dissolution (Barth et al., 2003. *Chemical Geology*, 200, 203-216;

Brunet et al., 2009. *Chemical Geology*, 265, 563-572). This $\delta^{13}C$ signature is significantly different from the observed $\delta^{13}C$ values of the emitted $CO_2$. Please also refer to our detailed responses to your major comment above. (lines 495-504; 525-528; 533-540)

L32: Define "NEP". I don't follow the last sentence of the abstract. What is meant by "…has been significantly offset by riverine carbon export"?
Reply: The definition of NEP 'net ecosystem production' has been inserted into the abstract. Because the lateral C export into the Wuding river network can be considered a loss of carbon from its terrestrial ecosystems, whether it is related to lateral transport of soil $CO_2$ (i.e., respiration taking place in soils) or lateral transport of SOC that is processed within the aquatic column. Therefore, the lateral transport of C from the upland terrestrial biosphere to the Wuding river network and its subsequent outgassing to the atmosphere offsets the estimates of terrestrial NEP. Similarly, Borges et al. (2015) discovered that riverine $CO_2$ evasion in African rivers offsets the terrestrial NEP of the Arica (Borges et al., 2015. *Nature Geoscience*, 8, 637-642). We have rephrased this claim in the revised text. (lines 32-34)

L38: "Rivers play an exceptionally significant role by directly linking…" Role in what? The global carbon cycle?
Reply: 'The global carbon cycle' has been added into the text.

L39: Remove dash between "terrestrially" and "derived".
Reply: Removed

L43: add "the" between "along" and "river".
Reply: Added.

L44: Remove comma after "processes".
Reply: Removed.

L45: Change "in-situ" to "in situ" for consistency.
Reply: Changed throughout the manuscript.

L46: How up-to-date is this 1.8 Pg C yr-1 number? See Drake et al. (2017) L&O Letters for an updated number.
Reply: Many thanks for your information. We have checked the updated C outgassing from global rivers and streams in Drake et al. (2017), which is now 3.2 Pg C year$^{-1}$, excluding the outgassing from non-running inland waters. This has been inserted into the revised manuscript. (lines 45-47)

L51: Has the number of studies on riverine carbon really been increasing exponentially? Change "recent" to "last".
Reply: To better describe the increasing studies on riverine carbon, we have reworded the statement: 'Although studies on riverine fluxes of carbon have been considerably increasing over the recent last 20 years…'. (lines 53-54)

L66: this should read "…through the drainage network to the catchment outlet…"

Reply: Revised. Many thanks.

L67: Remove "in" before "northern".
Reply: Removed.

L83: This should read "…and is located"
Reply: Added.

L85: Consider defining "loess" here.
Reply: Because we divided the river catchment into two subcatchments based on the geomorphological landscape. For the loess subcatchment (Figure 1), it is generally covered with 50–100 m deep loess soils. (lines 88-89)
.
L92: Citation for hydrologic regime description?
Reply: A reference (Li et al., 2007. Assessing the impact of climate variability and human activities on streamflow from the Wuding River basin in China. *Hydrological Processes*, 21, 3485-3491) has been inserted to support the statement. (lines 94-95)

L94: I'm confused – is this sentence saying that one particular extreme event led to an erosion rate of 7000 t km-2 yr-1 for a particular year? If so, what is the average erosion rate? I feel like this would be more informative.
Reply: The Wuding River catchment suffered severe soil erosion during the period 1956-1969, prior to the implementation of large-scale soil conservation programmes which were initiated from the early 1970s. The average soil erosion rate is about 7000 t $km^{-2}$ $yr^{-1}$ during this period. Since then, the soil erosion rate has been significantly reduced due to soil conservation, and current soil erosion is only 1500 t $km^{-2}$ $yr^{-1}$. We have reworded the description in the revised manuscript. (lines 95-97)

L102-103: "[The altered CO2 exchange] remains to be quantified". Didn't you quantify this in Ran et al. (2017)? If so, how does this "remain to be quantified"?
Reply: We have quantified the $CO_2$ exchange in our earlier work (i.e., Ran et al., 2017. *Journal of Geophysical Research: Biogeosciences*, 122, 1439-1455) and have removed this statement from the revised manuscript.

L110: Change "was" to "is".
Reply: Changed.

L124: Change "triple" to "triplicate".
Reply: Changed.

L127: "radiocarbon Δ14C samples" is somewhat redundant. Change to "collected samples for 14C analysis" or similar.
Reply: Changed. Many thanks.

L147 (and throughout): "The Δ14C values were reported as percent modern carbon (pMC)". These are two separate units! (see above discussion).

Reply: We have clarified the descript in the revised manuscript 'The [14]C results were reported as percent modern carbon (pMC)'. (lines 153-156)

L156: How reliable is this method for calculating carbon loads? This seems too simple. Why was something like LoadEst not used?
Reply: Estimating riverine carbon flux is a very important part of this study in which we attempt to investigate the fate of carbon after entering the drainage network from terrestrial ecosystems. Just as you have pointed out, there are a number of methods to estimate the annual fluxes of dissolved and particulate matter transported by rivers. Major methods currently used include linear interpolation and ratio estimators, regression-based methods historically employed by the USGS, and recent flexible techniques such as Weighted Regressions on Time, Discharge, and Season (WRTDS), etc. As you have also suggested, the most commonly used USGS software package for estimating constituent load using regression is known as LOADEST (Runkel et al., 2004. Load Estimator (LOADEST): A FORTRAN Program for Estimating Constituent Loads in Streams and Rivers. U. S. Geological Survey Techniques and Methods Book 4, Chapter A5). Lee et al. (2016) recently reviewed the potential for flux estimation bias across a broader range of estimation methods and concluded that the Beale's ratio estimator and WRTDS generally exhibit greater estimation accuracy and lower bias (Lee et al., 2016. An evaluation of methods for estimating decadal stream loads. *Journal of Hydrology*, 542, 185-203). Our annual carbon flux estimation in this study was based on the Beale's stratified ratio estimator. Since the riverine carbon concentrations were measured with "sparse" sampling frequency while flow and suspended sediment had a continuous daily measurement, this method could greatly reduce the bias introduced by relatively low sampling frequency, in particular the high flow events that are often under-sampled (Parks and Baker. 1997. Sources and transport of organic carbon in an Arizona river-reservoir system. *Water Research*, 31, 1751-1759). Indeed, we have already used the Beale's ratio estimator in our earlier estimation of carbon flux in the Yellow River with success (i.e., Ran et al., 2013. Spatial and seasonal variability of organic carbon transport in the Yellow River, China. *Journal of Hydrology*, 498, 76-88). And the Beale's ratio estimator has proven to be highly reliable and is recommended if the relationship between discharge and concentration is weak (e.g., Fulweiler and Nixon, 2005. *Biogeochemistry*, 74, 115-130; Awad et al., 2017. *Environmental Pollution*, 220, 788–796; Chen et al., 2014. *Journal of Geophysical Research: Biogeosciences*, 119, 95-109; Sun et al., 2017. *Hydrological Processes*, 31, 2062-2075). In comparison, we have also estimated the carbon flux by using the suggested LOADEST software package. The flux results show high consistency with each other, with a difference of less than 4.5%. We have added a detailed description of the estimate method (i.e., the Beale's ratio estimator) in the revised manuscript. Please refer to the highlighted changes in the text. (lines 161-180)

L160: "by multiplying annual sediment deposition rate…" How was deposition rate calculated? This is not described at all in the text.
Reply: Our earlier work (Ran et al., 2013. *Global and Planetary Change*, 100, 308-319) has estimated the average annual sediment deposition rate behind all check dams in the study catchment by considering sediment input into each check dam and its sediment trapping efficiency. This has been added into the revised manuscript. (lines 191-193)

L162: Change "was" to "were". Were these CO2 flux data taken directly from Ran et al. (2017),

or are these new data originally presented in this study? Overall, I would clearly state which data are new and which data are taken from previous studies (as these authors appear to have published multiple papers on this dataset…)

Reply: Changed. The $CO_2$ efflux data are taken from our earlier work (Ran et al., 2017). This study is built upon our earlier work (i.e., Ran et al., 2017). But in Ran et al. (2017), we only explored the environmental controls and dam impoundment impact on areal $CO_2$ emissions (mmol m$^{-2}$ d$^{-1}$). In this study, we aim to evaluate the riverine C budget by considering lateral C export, OC burial, and $CO_2$ emissions from the whole drainage network. We have clearly stated which data are new (thus presented in the manuscript or in Supplement) and which data have been presented in our earlier work (i.e., Ran et al., 2017) in the revised manuscript. (lines 146; 187-188; 23-238; 271-272; 309-310)

L172: Please provide the minimum catchment threshold area, as this will affect calculated Strahler stream order.

Reply: To delineate the stream network, a threshold value of 100 cells (90-m resolution) was set on the assumption that a stream initiates within the cells. The delineated stream network was then classified using the Strahler ordering system. We have applied this minimum catchment threshold to delineate the whole Yellow River catchment and the result was validated with ground-truthing (Ran et al., 2015. *JGR-Biogeosciences*, 120, 1334-1347). This description has been added into the manuscript. (lines 195-200)

L177: How valid is the assumption that "each round of field sampling [is] representative of CO2 emissions" for these four-month periods? What about for DIC, DOC, and POC concentrations – presumably you assume these are representative too?

Reply: Located in the arid-semiarid climate zone, the surface water $pCO_2$ in the Wuding River catchment shows temporal variations between the dry and wet seasons (please see Ran et al., 2017. *JGR-Biogeosciences*, 122, 1439-1455). However, the $pCO2$ is generally consistent within the dry (or wet) season, which is probably because of the dominance of groundwater inflow. Our fieldwork in each campaign lasted ~25 days, and repeated $pCO_2$ measurements at some sites over 20-day intervals show high consistency (e.g., <6% difference). Thus, in view of the hydrologic regime (mainly groundwater input), we assumed that the three sampling campaign results in different seasons are representative of $CO_2$ emissions in the three four-month periods. To make it clearer, we have added 'a first-order estimate' into the statement. As for the DIC, DOC, and POC concentrations, we instead used the monthly sampling results at the catchment outlet (i.e., Baijiachuan gauge in Fig. 1) for the yearly flux calculation. (lines 205-206)

L189: Heterotrophic soil respiration need not be due to bacteria – this could also be fungal or archaeal respiration. I would simply stick with "heterotrophs".

Reply: Many thanks. We have revised the statement and 'heterotrophs' was used instead.

L216: Is this decline (insofar as it is statistically significant) really "remarkable"?

Reply: We performed a one-way ANOVA test for the DIC in the loess subcatchment. The p value in spring, summer, and autumn is 0.02, 0.05, and 0.01. Thus, we concluded that this decline is remarkable (statistically significant). We have added the 'one-way ANOVA test, $p \leq 0.05$' into the manuscript to justify the claim. (lines 245-247)

L225: I'm confused by the sentence beginning with "Because the flow regime in 2017 was significantly biased…" What is this saying? You applied the 2015 hydrological regime to the 2017 data?

Reply: The flow regime in 2017 was significantly biased due to an extreme flood on 25-26 July caused by heavy rainstorms (maximum daily rainfall: 203 mm; spontaneous discharge: 4490 $m^3$/s with a return period of 200 years. Figure S1 in Supplement). In comparison, the multi-annual mean water discharge is 35 $m^3$/s. As a result, the annual water flux in 2017 is 1.5-fold the recent mean annual water flux (2000-2015). Because both $CO_2$ emissions and NPP were measured in 2015, we used the hydrological data for 2015 to calculate downstream carbon export by assuming that carbon concentration was comparable in 2015 and 2017 and evaluated the carbon budget. We realized that this may have caused errors to the flux estimation. We also calculated the carbon flux based on the 2017 flow data. The results show that, if the extreme flood on 25-26 July was excluded, the carbon flux in 2017 is close to that in 2015 ($7.3 \times 10^{10}$ vs. $(7 \pm 1.9) \times 10^{10}$ g). We have revised the statement in the manuscript for clarity and a new reference has been added to justify the impact of this extreme flood (He et al., 2018. *Geomatics, Natural Hazards and Risk*, 9, 70-18) (lines 256-261). In addition, we have also added a detailed description of the extreme flood event on 25-26 July in the Supplement (Figure S1).

L230: Fluxes should be in units of "g C yr-1" (I'm assuming the "yr-1" got dropped by accident).

Reply: Because we have already mentioned the 'annual' in the statement (The *annual* downstream carbon export…), adding 'yr-1' is redundant.

L257: This should read "Assuming the water surface area remained constant…"

Reply: Changed.

L265 (and throughout) Please add "VPDB" after "‰" when reporting d13C values.

Reply: Added throughout the manuscript. Thanks a lot.

L277: Should "soils" instead read "sediments"? How can sediment cores contain "soils"?

Reply: Revised.

L300: How turbid are these rivers? If they are quite turbid, then I would expect that photochemical degradation is probably insignificant.

Reply: The Wuding River catchment is one of the major sediment sources of the Yellow River as a result of severe soil erosion. The average suspended sediment concentration in recent years is in the range of 7900-11,000 mg/L, and it can reach 120,000 mg/L during floods. The extreme of in 2017 is recorded on 26 July at 488,000 mg/L. Thus, we have removed the claim of photochemical degradation from the text. (lines 340-342)

L301 (and throughout): Please change "labile" to "bioavailable" as this language is more consistent with our current understanding of OC decay dynamics.

Reply: Changed throughout the manuscript.

L312: I'm confused by the statement "…and the mixture of carbon export from the two subcatchments." I thought the "6th mainstem channels" are the two subcatchments, which combine to form the 7th order Wuding River? Or have I misinterpreted this? (It is hard to see on

Figure 1).

Reply: We have double checked the drainage network in Figure 1. The mainstem channel of the northwestern sandy subcatchment is in the 5th order and the mainstem channel of the southwestern loess subcatchment is in the 6th order. Thus, when the two subcatchment mainstem channels confluence, it is still a 6th order river. We have carefully adjusted this figure in terms of color scheme, marker size, label size, etc., and have added the subcatchment boundaries to make the figure much easier to read. Please refer to the revised Figure 1 for the changes.

L330: Similarly, please change "biogeochemically refractory" to "persistent" in order to be consistent with our current understanding of OC decay dynamics.

Reply: Many thanks for your suggestion. We have replaced this statement in the text.

L358 & 362: Phytoplankton are not aquatic plants. Please clarify this language.

Reply: Thanks a lot for your comment. While phytoplankton are plant-like in their ability to use sunlight to convert $CO_2$ and water into energy, they are not plants. We have reworded the claim in the revised manuscript '…intensive nutrient loading from agricultural fields may have facilitated the growth of phytoplankton like algae, …'. (lines 402-405)

L426: I'm confused by the inclusion of this sentence – what does it add to the discussion?

Reply: It seems this sentence is irrelevant to the discussion as you commented. It has thus been removed.

L464: Is the correlation between d13C and Δ14C statistically significant? Figure 9 does not report the regression slope equation nor any statistics, so I have no way of gauging the strength of this relationship (I'm not even sure if the line drawn in Figure 9 is a regression line…) Please clarify.

Reply: Based on your comment, we have performed the regression analysis by using the results of all the three sampling campaigns and have added the regression equation (slope and $r^2$) into the revised figure. Accordingly, we have clarified the argument in the revised text. Thanks a lot for your comment. (lines 525-528)

L470: "…which suggests the outgassing of ancient terrestrial OC after entering aquatic systems". I'm confused here – OC itself cannot be outgassed. Does this refer to CO2 generated from remineralization of old OC? If so, how do the authors know that this was remineralized after entering the aquatic system and not simply remineralized in soils and transported with soil pore waters?

Reply: As you have commented, the OC itself cannot be directly outgassed from the water-air interface. Here it refers to the $CO_2$ generated from remineralization of old OC, which is mineralized either in soils and then transported into rivers or in aquatic systems during transit. We have reworded the claim in the revised manuscript 'This suggests that the emitted $CO_2$ is derived from ancient terrestrial OC which is mineralized either in soils and then transported into rivers or in aquatic systems during transit…'. (lines 533-536)

L472: This claim (and others throughout the manuscript) isn't necessarily supported – I would urge caution when making concrete statements such as this. Rather, I would phrase this along the lines of "These results are consistent with…"

Reply: Thanks a lot for your comment. We have rephrased the wording of these claims, including this one and also others throughout the manuscript, to make them more accurate and appropriate. Thanks again. (lines 469-482; 495-504; 525-528; 533-540, etc.)

L510: I'm confused by the statement "…this percentage (16%) falls into the range of global-scale estimates of 50-70%..." 16% is not in the range of 50-70%... am I missing something?
Reply: Many thanks for pointing out this misinterpretation. The percentage of 16% in this study is lower than the global-scale estimate of 50-70% by Cole et al. (2007). We have revised this in the manuscript '… this percentage (i.e., 16%) is much lower than the global-scale estimate of 50   70% by Cole et al. (2007)'. (lines 575-577)

L514: In what way is the estimate of Cole et al. (2007) "conservative"?
Reply: Based on published estimates of gas exchange, sediment accumulation, and carbon transport, Cole et al. (2007) constructed a carbon budget for the role of inland waters (particularly lakes, rivers, and reservoirs) in the global carbon cycle. However, constrained by data availability, they were not able to characterize carbon transport in each inland water body and in most cases, they used mid-range values for the estimation. This can be seen from their paper, they repeatedly mentioned that their carbon flux estimates are conservative and associated with considerable uncertainties. The full citation is: Cole et al., 2007. Plumbing the global carbon cycle: Integrating inland waters into the terrestrial carbon budget. *Ecosystems*, 10, 171-184.

L516-517 (and 531): Please remove "… it is worth noting that".
Reply: Both have been removed based on your suggestions.

L537: Remove the dash between "terrestrially" and "derived".
Reply: Changed.

L548: "CO2 emissions represented an important pathway…" An important pathway for what? Carbon loss from the landscape? I would change this to "CO2 emissions are quantitatively important…" or similar.
Reply: We have rephrased the claim based on your suggestion. Thanks a lot.

Figure 1: This figure is hard to read given the current color scheme and marker sizes. I would consider changing the color scheme for clarity and making the markers significantly larger. Also, please provide a catchment outline for the "sandy" and "loess" subcatchments, as this delineation is currently unclear.
Reply: Thanks a lot for your suggestion. We have carefully adjusted this figure in terms of color scheme, marker size, label size, etc., and have added the subcatchment boundaries to make the figure much easier to read.

Figures 2-3: I would consider writing "Loess Subcatchment" and "Sandy Subcatchment" above panels (a) and (b) so that the reader does not have to dig through the caption to understand what is presented.
Reply: The names of the two subcatchments have been added into the two panels in both figures. Many thanks for your suggestion.

Figure 4: I'm confused by what the percentage numbers represent. This should be clarified in the figure caption.
Reply: The percentage above each order in (b) represents the proportion of $CO_2$ emissions from that order streams to the total $CO_2$ emissions. This has been clarified in the figure caption.

Figure 5: Why has the nomenclature and color scheme changed for this figure? Why not use the colored bars from Figures 2-3 and the "spring", "summer", "autumn" notation that is used throughout the text? Also, what is meant by "conventional age"? Is this equivalent to 14C yr BP?
Reply: The nomenclature and color scheme for this figure have been adjusted for consistency by using the same notation in the text. In addition, the caption has been revised to 'Seasonal variations in radiocarbon ages (year before present, BP) for the emitted $CO_2$ from the Wuding River catchment'.

Figure 7: Why is this figure showing NPP when the authors are interested in NEP? Why not show NEP directly? Also, please include the river network, subcatchment outlines, labels, etc. as in Figure 1.
Reply: Because NEP is calculated from NPP by subtracting the heterotrophic soil respiration ($R_h$) and the $R_h$ is just a single numerical figure, we presented the spatial variation of the NPP within the catchment. The river network, subcatchment outlines, labels, etc. have now been included in the revised figure.

Figure 9: Why are units of pMC (which is not the same as Δ14C!) used in this Figure but 14C years used in Figure 5? Is this dashed line a regression line? If so, please include the regression equation and statistics. Technically, "young" and "old" only correspond to the y-axis and should point vertically, as the x-axis of this figure says nothing about age.
Reply: Based on your earlier comment, we chosen to use the percent modern (pMC) to describe the [14]C results throughout the manuscript. To compare our results with Wang et al (2012. *Global Biogeochemical Cycles*, 10, 26, doi:10.1029/2011GB004130) that investigated the [14]C age of DOC and POC in the Yellow River, we kept the '[14]C yr BP' results in Figure 5. In addition, we have included the regression equation and statistics ($r^2$) in the figure.

Figure 10: I'm confused by the inset pie chart – what does 100% represent? Is this all of the carbon in the river network? If so, at what time points, or does this represent the relative annual fluxes? Again, more detail in the caption would be very much appreciated.
Reply: The inserted pie chart denotes the partitioning of riverine carbon among its five phases with the sum (100%) representing all the carbon entering the river network (i.e., $(18.5\pm4.5)\times10^{10}$ g C year$^{-1}$). We have added more details into the figure caption.

Figure captions: In general, I would like to see significantly more description in this figure captions.
Reply: We have significantly improved the figure captions on the basis of your comments and detailed information has been added. Please refer to the highlighted changes in the revised manuscript.

---

## Author Comment (AC3) · 16 Apr 2018

Dear Dr Lee,

Many thanks for your comments on our manuscript. Based on your very constructive comments, we have thoroughly revised the manuscript. Additional discussion and justifications have been added into the manuscript or into the Supplement. Please see below the detailed responses. Major changes have also been highlighted in the revised manuscript.

With best regards
Lishan Ran, on behalf of the coauthors
* * *
This study provides rich river carbon data from a watershed influenced by arid-semiarid climate. The data, including river carbon concentrations, exports, contents, and emissions in different carbon species, are very informative. I believe that more careful analyses of these comprehensive data can enhance our understanding of river carbon cycling and its role in linking terrestrial and marine biogeochemistry. I found some small and large problems which I think should be addressed for publication of this manuscript in Biogeosciences.

Estimation method of river carbon exports P4L156-160: River carbon exports are one of key results of this study, and thus should be estimated very carefully. However, I found that the estimate method of the exports is not clear. There are various estimation methods that could be applied. Aulenbach, B.T., Buxton, H.T., Battaglin, W.A., and Coupe, R.H., 2007, Streamflow and nutrient fluxes of the Mississippi-Atchafalaya River Basin and subbasins for the period of record through 2005: U.S. Geological Survey Open-File Report 2007-1080, https://toxics.usgs.gov/pubs/of-2007-1080/index.html Cohn, T.A., Caulder, D.L., Gilroy, E.J., Zynjuk, L.D., Summers, R.M., 1992, The validity of a simple statistical model for estimating fluvial constituent load-sâA˘T˘An empirical study involving nutrient loads entering Chesapeake Bay: Water Resources Research, v. 28, no. 9, p. 2353–2363. Runkel, R.L., Crawford, C.G., and Cohn, T.A., 2004, Load estimator (LOADEST)âA˘T˘A FORTRAN program for estimating constituent loads in streams and rivers: U.S. Geological Survey Techniques and Methods, book 4, chap. A5, 69 p.

Reply: Estimating riverine carbon flux is a very important part of this study in which we attempt to investigate the fate of carbon after entering the drainage network from terrestrial ecosystems. Just as you have pointed out, there are a number of methods to estimate the annual fluxes of dissolved and particulate matter transported by rivers. Major methods currently used include linear interpolation and ratio estimators, regression-based methods historically employed by the USGS, and recent flexible techniques such as Weighted Regressions on Time, Discharge, and Season (WRTDS), etc. As you have also suggested, the most commonly used USGS software package for estimating constituent load using regression is known as LOADEST (Runkel et al., 2004. Load Estimator (LOADEST): A FORTRAN Program for Estimating Constituent Loads in Streams and Rivers. U. S. Geological Survey Techniques and Methods Book 4, Chapter A5). Lee et al. (2016) recently reviewed the potential for flux estimation bias across a broader range of estimation methods and concluded that the Beale's ratio estimator and WRTDS generally exhibit greater estimation accuracy and lower bias (Lee et al., 2016. An evaluation of methods for estimating decadal stream loads. *Journal of Hydrology*, 542, 185-203). Our annual carbon flux estimation in this study was based on the Beale's stratified ratio estimator. Since the riverine carbon concentrations were measured with "sparse" sampling frequency while flow and

suspended sediment had a continuous daily measurement, this method could greatly reduce the bias introduced by relatively low sampling frequency, in particular the high flow events that are often undersampled (Parks and Baker. 1997. Sources and transport of organic carbon in an Arizona river-reservoir system. *Water Research*, 31, 1751-1759). Indeed, we have already used the Beale's ratio estimator in our earlier estimation of carbon flux in the Yellow River with success (i.e., Ran et al., 2013. Spatial and seasonal variability of organic carbon transport in the Yellow River, China. *Journal of Hydrology*, 498, 76-88). And the Beale's ratio estimator has proven to be highly reliable and is recommended if the relationship between discharge and concentration is weak (e.g., Fulweiler and Nixon, 2005. *Biogeochemistry*, 74, 115-130; Awad et al., 2017. *Environmental Pollution*, 220, 788–796; Chen et al., 2014. *Journal of Geophysical Research: Biogeosciences*, 119, 95-109; Sun et al., 2017. *Hydrological Processes*, 31, 2062-2075). In comparison, we have also estimated the carbon flux by using the LOADEST software package. The flux results show high consistency with each other, with a difference of less than 4.5%. We have added a detailed description of the estimate method (i.e., the Beale's ratio estimator) in the revised manuscript. Please refer to the highlighted changes in the text. (lines 161-180)

Estimation method and uncertainty of NEP P5L182-199 and P7281-291: For river carbon budget analysis, the NEP result is critical to drive the conclusion. However, I am a bit skeptical about the approach to calculate NEP. The authors are using different independent data sources for NPP and SR, and then, to calculate Rh, adapting another study's assumption "Rh accounts for 54% and 40% of SR in forested and non-forested areas,". This methodology probably led to large uncertainty in the final NEP estimate, which should be at least discussed.

Reply: Thanks a lot for your comment. Estimating NEP is quite important for the carbon budget analysis of this study. We divided the study catchment into two subcatchments, including the sandy subcathment and the loess subcatchment. While forest cover in the Wuding River catchment is quite low (less than 5%) as a result of low precipitation, grassland is the major land cover in the sandy subcatchment and agriculture and grassland predominate the loess subcatchment (Wang et al., 2014. Spatial-temporal changes of land use in Wuding River Basin under ecological restoration, *Bulletin of Soil and Water Conservation*, 34, 237-243 (in Chinese with English abstract). This is largely the result of the implementation of the Grain-for-Green Project which was initiated by the Chinese government in 1999. After more than 10 years of implementation of this vegetation restoration program, the vegetation cover (forest and grassland) has greatly increased. Please also refer to two photos below showing the landscape of the sandy subcatchment (left) and of the loess subcatchment (right). Both photos were taken by me in 2015 during the fieldwork). To better describe the landscape of the catchment, we have revised the description in Section 2.1 'Study area' (lines 88-90). Therefore, the landscape and land cover of the Wuding River catchment are generally consistent with the distinction of "forested" and "non-forested" by Hanson et al. (2000). With respect to the huge range of the "non-forested" fraction heterotrophic (i.e.,10-90%), we have discussed the potential uncertainty in the revised manuscript. (lines 560-563).

Our rate is consistent with recent measurements under different vegetation types in this arid-semiarid region (e.g., Fu et al., 2013). Fu et al. (2013. Soil respiration as affected by vegetation types in a semiarid region of China. *Soil Science and Plant Nutrition*, 59, 715-726) measured total soil respiration in this arid-semiarid region. Their mean soil respiration rates under 4

different vegetation types are in the range of 1-1.4 µmol m$^{-2}$ s$^{-1}$, which are equivalent to 380-530 g C m$^{-2}$ year$^{-1}$. Thus, our estimate is reliable. We have carefully revised the manuscript with new references to justify our arguments (lines 316-326). Using the ratios derived from Hanson et al. (2000) has been widely used in the world to assess heterotrophic soil respiration in river catchments under different land cover types (e.g., Brunet et al., 2009. Terrestrial and fluvial carbon fluxes in a tropical watershed: Nyong basin, Cameroon. *Chemical Geology*, 3, 563-572; Lee et al., 2017. A high-resolution carbon balance in a small temperate catchment: Insights from the Schwabach River, Germany. *Applied Geochemistry*, 85, 86-96). Just as you have commented, this portioning is associated with potential uncertainty. We have further discussed this in the revised manuscript. (lines 320-326; 560-563)

[Figure]

Figure: Landscape characteristics of the sandy (left) and loess (right) subcatchments.

Data availability and clarification A strength of this study is that it provides and interpret the very comprehensive river carbon data. Biogeosciences readers would be interested to see the data/results in more detail. There are many results which are described in texts, yet cannot be directly read by figures or tables. Also, the authors might want to have a simple table that lists the data with time (which year, season,...), units (concentration, contents, exports...), and brief estimation methods. This study covers a lot of interesting data, but I am confused by how they were presented. Also, I am confused by the use of "concentrations" and "contents".
Reply: Based on your and other reviewers' comments, we have compiled all the data that are not included in our earlier work (i.e., Ran et al., 2017. *Journal of Geophysical Research: Biogeosciences*, 122, 1439-1455) in the Supplementary Information. In the Supplementary Information of the Ran et al. (2017) paper, we have already made most of our raw data used in this study available. These data include the physiochemical parameters (e.g., sampling time/season, location, elevation, channel slope, flow velocity, wind speed, pH, water temperature, dissolved oxygen, Chl *a*, etc.), $CO_2$ emissions ($pCO_2$ and areal flux), and dissolved carbon concentration (DOC and DIC) in both river and reservoir waters. To facilitate future review studies and/or comparison analyses, we have made the leftover data available by presenting them in the Supplementary of this study. Specifically, these data include POC of sediment samples (2015 and 2017) and of drilled sediment from check dams (2015), monthly DOC and DIC concentrations at the catchment outlet (Baijiachuan gauge, 2017) as well as the concomitant flow information. Please refer to the Supplement for these data (Tables S1-S3).

P1L15: What do you mean by "redistribution"?
Reply: Here we meant the fate of riverine carbon during its transport from headwater streams to the catchment outlet, including downstream export to catchment outlet, $CO_2$ evasion from water

surface, and organic carbon (OC) burial through sediment storage. We have replaced the word 'redistribution' with 'fate' for clarity in the text. (lines 15 and 69)

P1L17-18: I am not sure what you meant with this "While the DOC concentration was spatially comparable within the catchment," I would remove this.
Reply: Based on your comment, we have removed this ambiguous claim and rephrased the abstract. (lines 17-20). Many thanks.

P1L18-19 vs. P8L312-314: Is this sentence consistent with your claims in P8L312-314? I am confused. "it was generally higher in spring and summer than in autumn, especially in the loess subcatchment." vs. "There was no discernible seasonal difference in DOC concentrations in both subcatchments, although the hydrograph varied significantly among the three seasons."
Reply: Many thanks for your comment. We have reworded these inconsistent arguments in the text. The DOC concentration showed no significant seasonal differences among the three sampling campaigns and was not sensitive to flow dynamics, although the flow discharge changed by a factor of 3. This likely reflects the predominance of groundwater input over the entire year and its highly stable DOC, which may have masked the 'dilution effect' with lower DOC concentrations usually observed in high-flow periods. Please refer to the highlighted changes in the manuscript. (lines 17-20; 351-355; 605-609)

P1L19-21 vs P8314-321 vs P9L375-377: I am also confused that these discussions appear to contradict each other. High soil carbon leaching due to high rainfalls in many cases leads to high river carbon exports (massC/time), but not high river carbon concentrations (massC/volume H2O). High rainfalls increase river flows as well, so concentrations can increase or decrease.
Reply: We completely agreed with your comments. High soil organic carbon leaching due to high rainfalls tends to result in high riverine carbon export (mass C), but not high DOC or POC concentrations (mg/L or POC% in suspended solids). This largely reflects the 'dilution effect' during high-flow periods, especially in (sub)tropical and temperate catchments with continuous surface runoff contribution in the wet season. In the arid-semiarid Wuding River catchment, although there were no significant seasonal differences in the riverine carbon concentrations (massC/water volume) between the three sampling campaigns, the carbon fluxes in the wet season (high-flow periods) were much higher than that in the dry season. This can also be discovered from the annual carbon flux at the catchment outlet estimated from monthly sampling. Based on your comments, we have carefully revised these claims in the manuscript. (lines 17-21; 417-429; 434-438; 442-447)

P1L23 and P5L209: Did you mean "showed" by "shown"?
Reply: Changed.

P2L84, P2L89, P2L94: An exact time period or years should be provided.
Reply: The time periods of mean water discharge (1956–2007), annual precipitation (1956–2004), and soil erosion (1956–1969) have been added into the revised manuscript. (lines )

P6L225-228: The assumption should be justified better. Why did you particularly use hydrological data for 2015?

Reply: The flow regime in 2017 was significantly biased due to an extreme flood on 25-26 July caused by heavy rainstorms (maximum daily rainfall: 203 mm; spontaneous discharge: 4490 m3/s with a return period of 200 years. Figure S1 in Supplement). In comparison, the multi-annual mean water discharge is 35 $m^3$/s. As a result, the annual water flux in 2017 is 1.5-fold the recent mean annual water flux (2000-2015). Because both $CO_2$ emissions and NPP were measured in 2015, we used the hydrological data for 2015 to calculate downstream carbon export by assuming that carbon concentration was comparable in 2015 and 2017 and evaluated the carbon budget. We realized that this may have caused errors to the flux estimation. We also calculated the carbon flux based on the 2017 flow data. The results show that, if the extreme flood on 25-26 July was excluded, the carbon flux in 2017 is close to that in 2015 ($7.3\times10^{10}$ vs. $7\times10^{10}$ g). We have revised the statement in the manuscript for clarity and a new reference has been added to justify the impact of this extreme flood (He et al., 2018. G*eomatics, Natural Hazards and Risk*, 9, 70-18) (lines 256-261). In addition, we have also added a detailed description of the extreme flood event on 25-26 July 2017 in the Supplement (Figure S1).

P7L298: Did you mean "concentrations" by "contents"?
Reply: To make it clearer, we have revised the term and now use the 'POC content (POC%) in sediments' throughout the manuscript. Please also refer to our response to your comment below P8L326-328.

P7L299: Specify by providing values to support "both DOC and POC contents in the Wuding catchment were relatively low compared with most rivers in the world."
Reply: For the Wuding catchment, its DOC concentrations are comparable to the global average DOC of 5.4 mg/L while its POC% is lower than most rivers in the world (mean: 0.95%; Ludwig et al., 1996. *Global Biogeochemical Cycles*, 10, 23-41). These global averages have been inserted into the text. (lines 333-335)

P8L303: I am not sure if this statement is valid. "This decomposition is generally associated with increasing water residence time for bacterial respiration in downstream streams due to decreasing flow velocities." I don't think that flow velocity generally decreases toward downstream. I think that travel time generally increases toward downstream and longer travel times provide more opportunity for decomposition.
Reply: Thanks a lot for the comment. We completely agree with your explanation on the downstream decrease in organic carbon concentrations after checking the flow velocity changes along the stream order. Thus, we have revised this claim: 'This mineralization is generally associated with increasing water residence time for bacterial respiration in downstream streams due to longer travel times which increase the potential for in-stream processes on DOC'. (lines 340-342).

P8L326-328: I don't understand what you mean here.
Reply: For POC, we used the POC content (POC%) in the total suspended solids (TSS) to present the results. This is because we tried to compare our results in the Wuding River catchment with the POC% values of the global rivers. Ludwig et al. (1996. *Global Biogeochemical Cycles*, 10, 23-41) synthesized global POC export into the oceans by continental erosion via major rivers. The average POC% values of the global rivers vary from 0.3% to 10.1%, although values above 1.5% are only observed in rivers with very low suspended

sediment concentrations (i.e., <300 mg/L). In addition, Meybeck (1993) assumed that riverine suspended loads have an ancient sedimentary OC origin of about 0.5% on average (Meybeck, 1993. C, N, P and S in rivers: from sources to global inputs, in: Interactions of C, N, P and S Biogeochemical Cycles and Global Change, Edited by: Wollast, R., Mackenzie, F. T., and Chou, L., Springer-Verlag, Berlin, 163-193). Comparing the POC% in the Wuding River basin with that of the global rivers shows the POC% in the Wuding River is at the lower end of the global rivers, which reflects the ancient sedimentary OC origin of about 0.5% for fluvial sediments. We have revised our justifications with new references in the revised version. (lines 333-335; 366-368)

---

## Referee Report (RR1)

*Review of Ran et al. "Riverine carbon export in the arid-semiarid Wuding River catchment on the Chinese Loess Plateau" (bg-2018-51), first revision*

The authors have addressed the majority of my earlier comments and suggestions. Therefore, I now have only a handful of minor comments that I feel should be addressed before publication. Again, please do not hesitate to contact me for further discussion regarding this review.

Sincerely,

Jordon Hemingway
jordon_hemingway@fas.harvard.edu

**Minor Comments**

L7 (and throughout, including Table 1, Table 2, and Fig. 10): The significant figure in the tenth's place appears to be dropped, presumably when ending in a zero. For example, "7±1.9" on L7 should read "7.0±1.9", etc. Please update the significant figures to be consistent throughout.

L33: I'm still slightly confused about how lateral transport is "significantly offsetting" NEP. Perhaps re-word to something along the lines of: "It appears that a significant fraction of terrestrial NEP in this arid-semiarid catchment is laterally transported from the terrestrial biosphere to the drainage network." (or similar)

L70: It's not immediately clear what the "three pathways" is referring to. Consider re-wording to: "…among its three pathways; that is 1) downstream export to the catchment outlet, 2) $CO_2$ evasion from the water surface, and 3) organic carbon burial…"

L87: Insert a comma before "generally" and after "soils".

L96: Saying "once suffered" sounds like a single event, while the time period 1956–1969 implies a sustained phenomenon. Consider re-wording to something like: "… the Wuding River catchment has experienced a maximum, decadal averaged soil erosion rate as high as 7000 t km$^{-2}$ yr$^{-1}$ (1956-1969)" or similar.

L143: Were these Gran titrations or end-point titrations? This should be specified.

L149: Add "and pestle" after "mortar".

L157: Beta Analytic measures $\delta^{13}C$ using an off-line IRMS, not simultaneously on the AMS (AMS-derived $^{13}C$ compositions are generally neither precise nor accurate). See: https://www.radiocarbon.com/dietary-isotopic-analysis.htm

L169 (and 176): Remove the comma after "where".

L174: Add a line that says "and" between these two equations.

L222: Change "calculated" to "calculate".

L240: Change "averaged" to "average".

L290: I'm a bit confused by these sentences. I think the authors are saying that spring and autumn $CO_2$ outgassing fluxes summed to 246 million mol, summer *in*gassing flux was 208 million mol, and these add up to a *net out*gassing flux of 38 million mol. Then, when added with the *river* efflux estimate, the *catchment total* adds up to $(3.7\pm0.5)\times10^{10}$ g C in the year 2015. I would re-word these sentences to clarify this. Additionally, the reservoir $CO_2$ emissions estimates appear to have large uncertainties, which should be reported and addressed here. For example, I calculate the net outgassing flux to be $38\pm280$ million mol, which is, of course, indistinguishable from zero. Propagating this error, I calculate a *catchment total* value of $(3.7\pm0.6)\times10^{10}$ g C (note the higher uncertainty).

L299: There appears to be a typo in reporting these numbers (e.g. "-30.2±‰").

L300: "conventional" should be replaced by "radiocarbon"

L301 (and throughout, including Table 2 and Fig. 5): "years" should be "$^{14}$C yr BP"

L359: "leached" implies going from the solid to liquid phase. Consider changing this to "adsorbed within deeper soils…"

L475: I would recommend noting the possibility that secondary OC sources (namely, phytoplankton) could contribute to that observed in check-dam sediments.

L495: A $\delta^{13}C$ value of 0‰ for carbonate-dominated rivers is the DIC value, not the $CO_2$ value. Keep in mind that $CO_2$ will be more depleted than DIC.

L561: I feel that this could be expanded a bit. Specifically, how would the uncertainty in $S_R$ and $R_h$ propagate in to the estimated percent of NEP that is laterally exported? It would be useful to know how certain the authors are in their "16% of NEP" number. There appears to be uncertainty about the NEP number as reported in Fig. 10, but this isn't included in the text.

L604: I would be careful in claiming that this is a "typical" study area. It is likely quite unique due to its location on the loess plateau as well as the large anthropogenic disturbance (including check dams).

Table 1: What is the timescale for the "million mol" columns? Is this "million mol $CO_2$ yr$^{-1}$"?

Fig. 7: This is a lot of significant figures for the NPP legend! Do the authors really trust these values to be so precise?

Fig. 9: In light of the authors' response to my earlier comments, I think it might actually make sense to keep this figure in $^{14}$C yr BP rather than pMC. The authors stated that they wanted to use $^{14}$C yr BP in this study in order to compare to previous studies, which makes sense and, for consistency, it would be logical to report this figure in the same units. Note that, in that case, a linear regression between $^{14}$C yr and $\delta^{13}$C likely doesn't make much sense and could be removed. This seems okay to me since the authors don't really need (or discuss) this regression trend.

---

## Author Response (AR2)

Dear Dr. Abril,

Based on the two reviewers' valuable comments, we have thoroughly revised the manuscript. Please see below the detailed responses. All suggested changes to wording have been incorporated into the revised manuscript. Major changes have also been highlighted.

Many thanks for accepting our manuscript for publication in BG.

Best regards
Lishan Ran, on behalf of all co-authors
* * *
**Referee #1**

The manuscript is greatly improved with answers to my questions. I think this manuscript can be published with small corrections (see below).

The use of the terms, "concentration" and "content" is still confusing. For example, line 23: "POC%" is concentration not content. Please double check the use of the words throughout the manuscript.
Reply: We measured the bulk POC content of the total suspended solids (TSS). According to Tolhurst et al. (2005. Content versus concentration: Effects of units on measuring the biogeochemical properties of soft sediments. *Estuarine, Coastal and Shelf Science*, 63, 665-673), the *content* denotes the mass fraction or mass per unit mass and is a unitless ratio. Therefore, here the POC%, a percentage of the TSS (dry weight), is content. Similar expression can also be found in literature (e.g., Alin et al., 2008. Biogeochemical characterization of carbon sources in the Strickland and Fly rivers, Papua New Guinea. *Journal of Geophysical Research*, 113, F01S05). To make it clear and consistent throughout the manuscript, we have used the term 'POC%' throughout the manuscript.

Lines 45-: The range of CO2 emissions needs to be expanded to be conservative. Please check out "Lauerwald et al., 2015, Global Biogeochemical Cycles, Spatial patterns in CO2 evasion from the global river network".
Reply: The recent estimate of global $CO_2$ emissions from rivers of 0.65 Pg C year$^{-1}$ by Lauerwald et al. (2015) has been added into the text. Many thanks for your constructive suggestion.

**Referee #2**

Review of Ran et al. "Riverine carbon export in the arid-semiarid Wuding River catchment on the Chinese Loess Plateau" (bg-2018-51), first revision

The authors have addressed the majority of my earlier comments and suggestions.
Therefore, I now have only a handful of minor comments that I feel should be addressed before

publication. Again, please do not hesitate to contact me for further discussion regarding this review.

Sincerely,

Jordon Hemingway
jordon_hemingway@fas.harvard.edu

Dear Dr. Hemingway,

Many thanks for your insightful comments and suggestions, which have greatly improved our manuscript. Please find below our responses to each of your comments.

**Minor Comments**

L7 (and throughout, including Table 1, Table 2, and Fig. 10): The significant figure in the tenth's place appears to be dropped, presumably when ending in a zero. For example, "7±1.9" on L7 should read "7.0±1.9", etc. Please update the significant figures to be consistent throughout.
Reply: This is because all these figures end in a zero. We have updated all the figures to be consistent throughout the manuscript. Thanks a lot.

L33: I'm still slightly confused about how lateral transport is "significantly offsetting" NEP. Perhaps re-word to something along the lines of: "It appears that a significant fraction of terrestrial NEP in this arid-semiarid catchment is laterally transported from the terrestrial biosphere to the drainage network." (or similar)
Reply: Based on your suggestion, this sentence has been re-worded (lines 32-34). Many thanks.

L70: It's not immediately clear what the "three pathways" is referring to. Consider re-wording to: "…among its three pathways; that is 1) downstream export to the catchment outlet, 2) CO2 evasion from the water surface, and 3) organic carbon burial…"
Reply: Based on your suggestion, this sentence has been re-worded. (lines 70-73)

L87: Insert a comma before "generally" and after "soils".
Reply: Added.

L96: Saying "once suffered" sounds like a single event, while the time period 1956–1969 implies a sustained phenomenon. Consider re-wording to something like: "… the Wuding River catchment has experienced a maximum, decadal averaged soil erosion rate as high as 7000 t km-2 yr-1 (1956-1969)" or similar.
Reply: Based on your suggestion, this sentence has been re-worded. (lines 96-97)

L143: Were these Gran titrations or end-point titrations? This should be specified.
Reply: The alkalinity was determined by triplicate end-point titrations by using methyl orange as the indicator. This has been specified in the text.

L149: Add "and pestle" after "mortar".

Reply: Added.

L157: Beta Analytic measures d13C using an off-line IRMS, not simultaneously on the AMS (AMS-derived 13C compositions are generally neither precise nor accurate). See: https://www.radiocarbon.com/dietary-isotopic-analysis.htm
Reply: The 'isotope ratio mass spectrometer (IRMS)' has been added into the manuscript. Many thanks for providing very accurate details. (line 157)

L169 (and 176): Remove the comma after "where".
Reply: Removed.

L174: Add a line that says "and" between these two equations.
Reply: Added.

L222: Change "calculated" to "calculate".
Reply: Changed.

L240: Change "averaged" to "average".
Reply: Changed.

L290: I'm a bit confused by these sentences. I think the authors are saying that spring and autumn CO2 outgassing fluxes summed to 246 million mol, summer ingassing flux was 208 million mol, and these add up to a *net outgassing* flux of 38 million mol. Then, when added with the river efflux estimate, the *catchment total* adds up to $(3.7\pm0.5)\text{x}10^{10}$ g C in the year 2015. I would re-word these sentences to clarify this. Additionally, the reservoir CO2 emissions estimates appear to have large uncertainties, which should be reported and addressed here. For example, I calculate the net outgassing flux to be $38\pm280$ million mol, which is, of course, indistinguishable from zero. Propagating this error, I calculate a catchment total value of $(3.7\pm0.6)\text{x}10^{10}$ g C (note the higher uncertainty).
Reply: Just as you have mentioned, the spring and autumn $CO_2$ outgassing fluxes were summed to 246 million mol (i.e., 81 million mol in spring and 165 million mol in autumn; Table 1), the summer $CO_2$ ingassing flux was 208 million mol, and these added up to a net outgassing flux of 38 million mol. Based on your suggestion, we have re-worded these sentences to make them clearer. While for the reservoir $CO_2$ emissions estimates, the great uncertainties as shown in Table 2 were largely because $CO_2$ effluxes in reservoirs were characterized by large temporal variations, particularly in spring. While the sandy subcatchment reservoirs in spring showed net $CO_2$ outgassing (28 mmol m$^{-2}$ d$^{-1}$), the loess subcatchment reservoirs in spring act as a net $CO_2$ sink (-2.9 mmol m$^{-2}$ d$^{-1}$). This caused the great uncertainties in the $CO_2$ efflux estimate in spring, which propagated to a great uncertainty in the annual total $CO_2$ efflux estimate as you pointed out. However, because the $CO_2$ efflux from reservoirs accounted for only about 1.37% of the river $CO_2$ efflux estimate $(3.65\pm0.5)\times10^{10}$ g C) or 1.35% of the total $CO_2$ efflux (rivers + reservoirs), this uncertainty is not likely to significantly affect the catchment total $CO_2$ efflux estimate. Furthermore, we have updated the uncertainty of the total $CO_2$ efflux estimate (i.e., $(3.7\pm0.6)\times10^{10}$ g C). Thanks a lot for your valuable comments and suggestions. (lines 291-297)

L299: There appears to be a typo in reporting these numbers (e.g. "-30.2±‰").

Reply: These numbers have been updated.

L300: "conventional" should be replaced by "radiocarbon"
Reply: Changed.

L301 (and throughout, including Table 2 and Fig. 5): "years" should be "14C yr BP"
Reply: Changed throughout the manuscript.

L359: "leached" implies going from the solid to liquid phase. Consider changing this to "adsorbed within deeper soils…"
Reply: Changed.

L475: I would recommend noting the possibility that secondary OC sources (namely, phytoplankton) could contribute to that observed in check-dam sediments.
Reply: Many thanks for your insightful comment. We have added the possible contribution from secondary OC sources (e.g., phytoplankton) into the text. (lines 480-482)

L495: A d13C value of 0‰ for carbonate-dominated rivers is the DIC value, not the CO2 value. Keep in mind that CO2 will be more depleted than DIC.
Reply: Yes, the $\delta^{13}C$ value of 0‰ for carbonate-dominated rivers is the dissolved inorganic carbon value, not the emitted $CO_2$ value. Affected by the fractionation process (preferential outgassing of $^{12}CO_2$), the $\delta^{13}C$ value of the emitted $CO_2$ is more depleted than DIC. We have discussed this in the manuscript (lines 520-523).

L561: I feel that this could be expanded a bit. Specifically, how would the uncertainty in SR and Rh propagate in to the estimated percent of NEP that is laterally exported? It would be useful to know how certain the authors are in their "16% of NEP" number. There appears to be uncertainty about the NEP number as reported in Fig. 10, but this isn't included in the text.
Reply: Considering the land cover and land use type in the Wuding River catchment, we further evaluated the uncertainty associated with the $S_R$ and $R_h$ estimation: "If the ratio is reduced to 35%, the proportion of lateral export to NEP would decrease by 5.6%. Further research involving field experiments and remote sensing technique is thus needed to constrain this estimate". Based on your suggestion, we have expanded the uncertainty discussion, including the NEP results in Fig. 10, into the manuscript. (lines 564-569)

L604: I would be careful in claiming that this is a "typical" study area. It is likely quite unique due to its location on the loess plateau as well as the large anthropogenic disturbance (including check dams).
Reply: Revised. Thanks a lot.

Table 1: What is the timescale for the "million mol" columns? Is this "million mol CO2 yr-1"?
Reply: The timescale is year$^{-1}$. The unite has now been revised to 'million mol $CO_2$ yr$^{-1}$'.

Fig. 7: This is a lot of significant figures for the NPP legend! Do the authors really trust these values to be so precise?

Reply: Based on your comment, we have revised the legend by reducing the figures to 3 decimal places. Thanks a lot.

Fig. 9: In light of the authors' response to my earlier comments, I think it might actually make sense to keep this figure in 14C yr BP rather than pMC. The authors stated that they wanted to use 14C yr BP in this study in order to compare to previous studies, which makes sense and, for consistency, it would be logical to report this figure in the same units. Note that, in that case, a linear regression between 14C yr and d13C likely doesn't make much sense and could be removed. This seems okay to me since the authors don't really need (or discuss) this regression trend.

Reply: Based on your suggestion, we have re-plotted this figure in $^{14}$C year BP against $\delta^{13}$C for comparison with previous studies. Also, this figure is now consistent with Fig. 5 in which the same units (i.e., $^{14}$C year BP) were used. In addition, because we have discussed the relationship between radiocarbon age and $\delta^{13}$C in section 4.3, we choose to retain the linear regression trend as you suggested in your earlier comments. Many thanks for your very insightful and valuable comments and suggestions.

---

## Author Response (AR3)

Comments to the Author:

Dear authors

thanks for this revised version. This makes your MS an important contribution to BG. Before sending your manuscript for publication, please address this final comment:

« Assuming the water surface area remained constant (i.e., no significant seasonal fluctuations), the spring and autumn CO2 effluxes were summed to 246 million mol and the summer CO2 efflux was -208 million mol (Table 1). These added up to an annual net CO2 efflux of 38 million mol (or $0.05 \times 1010$ g C) with great uncertainties due largely to the spatial variation between the sandy and loess subcatchment reservoirs in spring (Table 1). When added with the river efflux estimate, the catchment total CO2 efflux was $(3.7 \pm 0.6) \times 1010$ g C in the year 2015."

In accordance with Dr Hemingway comments, insert here the idea that statistically, such small residual flux calculated as the difference between two large numbers with great uncertainties is probably not different from zero. The reservoir annual CO2 flux is closed to balanced and accounts for less than x% of total CO2 outgassing, mostly by rivers.

Thanks for submitting your work to BG

with best regards,

Gwenaël Abril
* * *
Dear Dr. Gwenaël Abril,

Many thanks for your comments on our manuscript. Based on your suggestion, we have corrected the manuscript. Now it reads "Assuming the water surface area remained constant (i.e., no significant seasonal fluctuations), the spring and autumn $CO_2$ effluxes were summed to 246 million mol and the summer $CO_2$ efflux was -208 million mol (Table 1). These added up to an annual net $CO_2$ efflux of $38 \pm 280$ million mol (or $0.05 \times 10^{10}$ g C), which is statistically indistinguishable from zero due largely to the spatial variation between the sandy and loess subcatchment reservoirs in spring (Table 1). When added with the river efflux estimate, the catchment total $CO_2$ efflux was $(3.7 \pm 0.6) \times 10^{10}$ g C in the year 2015, of which the reservoir $CO_2$ efflux accounted for less than 1.4%."

We greatly appreciate your and the three reviewers' valuable comments which have significantly improved our manuscript. Thanks a lot for accepting our manuscript for publication in BG.

Best regards
Lishan, on behalf of all co-authors